# Regulatory sequence-based discovery of anti-defense genes in archaeal viruses

Yuvaraj Bhoobalan-Chitty [1,3] ✉, Shuanshuan Xu[1,3], Laura Martinez-Alvarez [1,3], Svetlana Karamycheva[2], Kira S. Makarova [2], Eugene V. Koonin [2] & Xu Peng [1] ✉

In silico identification of viral anti-CRISPR proteins (Acrs) has relied largely on the guilt-by-association method using known Acrs or anti-CRISPR associated proteins (Acas) as the bait. However, the low number and limited spread of the characterized archaeal Acrs and Aca hinders our ability to identify Acrs using guilt-by-association. Here, based on the observation that the few characterized archaeal Acrs and Aca are transcribed immediately post viral infection, we hypothesize that these genes, and many other unidentified anti-defense genes (ADG), are under the control of conserved regulatory sequences including a strong promoter, which can be used to predict anti-defense genes in archaeal viruses. Using this consensus sequence based method, we identify 354 potential ADGs in 57 archaeal viruses and 6 metagenome-assembled genomes. Experimental validation identified a CRISPR subtype I-A inhibitor and the first virally encoded inhibitor of an archaeal toxin-antitoxin based immune system. We also identify regulatory proteins potentially akin to Acas that can facilitate further identification of ADGs combined with the guilt-by-association approach. These results demonstrate the potential of regulatory sequence analysis for extensive identification of ADGs in viruses of archaea and bacteria.

Prokaryotes encode a diverse range of innate and adaptive defense mechanisms including receptor modification, abortive infection, restriction modification systems, CRISPR-Cas, and many other systems that have been characterized recently[1]. The virus-host arms race leads to continuous expansion and diversification of immune systems in bacteria and archaea and concomitant evolution of anti-defense mechanisms in viruses and other mobile genetic elements (MGEs). MGEs have been long known to encode anti-defense proteins that inhibit restriction modification[2,3] and abortive infection[4–6] systems. More recently, diverse experimental strategies and the guilt-by-association bioinformatic approach have been used to identify numerous anti-CRISPR (Acr) proteins encoded by bacterial viruses[7–11].

The study of archaeal defense mechanisms has been more limited, with only surface resistance[12,13], argonaute[14–17], restriction-modification[18–20] and diverse CRISPR-Cas[21–23] characterized so far.

Moreover, despite the near ubiquity of CRISPR-Cas systems in archaea, only four Acrs have been experimentally identified in archaeal viruses, in contrast to more than 100 Acrs identified in bacteriophages[24–26]. The guilt-by-association approach so far found little application for archaeal viruses, primarily because the bait is typically surrounded by multiple monocistronic ORFs of unknown function[27]. The relatively small number of sequenced viral genomes, the lack of diverse genetic tools and the comparatively poor understanding of the fundamental biology of archaeal viruses add to the difficulty of identifying Acrs. The existence of multiple functionally uncharacterized paralogs of Acrs in a single archaeal viral genome[28] adds another layer of complexity.

Early expression of Acrs is crucial for the inactivation of the constantly expressed host CRISPR-Cas systems. Despite the early Acr expression, a substantial fraction of the infecting virus particles are

[1]Department of Biology, University of Copenhagen, Copenhagen N, Denmark. [2]National Center for Biotechnology Information, National Library of Medicine, NIH, Bethesda, MD, USA. [3]These authors contributed equally: Yuvaraj Bhoobalan-Chitty, Shuanshuan Xu, Laura Martinez-Alvarez. ✉e-mail: yuvarajb@bio.ku.dk; peng@bio.ku.dk

destroyed before immunosuppression is established. Avoidance of complete virus eradication depends on the early expression of Acrs and efficacy of their inhibitory activity, which together determine the tipping point, i.e. the critical threshold of the initial viral density[29–31]. Recently, it has been shown that the burst size also plays a role in the overall fate of viral infections[32]. The steep early expression of Acrs is followed by repression by the co-expressed Acr associated (Aca) protein[31], such that the resources of the immunosuppressed host are redirected in an orderly manner towards the expression of viral genes required for the completion of the viral life cycle. Hence, apart from encoding inhibitors of the host defense mechanism, it is essential for viruses to precisely regulate the timing of their gene expression.

Steep early expression of inhibitors appears to be crucial for a virus to counteract host defense systems that attack the virus in the early stage of its life cycle. Therefore, it appears likely that most early viral genes are anti-defense genes (ADGs). Here, we demonstrate that all the early genes of Icerudivirus SIRV2 (SIRV2), including *acrID1*[28] and *acrIIIB1*[33], share a highly conserved regulatory sequence surrounding a nearly identical promoter core (TATA box and BRE). By analyzing the available archaeal viral genomes, we identified 354 novel ADGs, possessing consensus regulatory sequences including the early promoter core, from 57 genomes of archaeal viruses and 6 metagenome-assembled sequences. These novel ADGs are classified into 116 families. As a proof-of-concept, an inhibitor of subtype I-A CRISPR-Cas was identified by screening the predicted ADGs of the lytic archaeal virus Sulfolobus islandicus filamentous virus 2 (SIFV2). Furthermore, the first example of an inhibitor of an archaeal toxin-antitoxin immune system was identified in several archaeal viruses including Sulfolobus monocaudavirus SMV1. Our results also suggest that transcriptional regulators previously defined as Acr associated, i.e. Aca, might regulate the expression of other anti-defense genes in addition to *acrs*.

## Results

### A consensus regulatory sequence precedes early expressed viral genes

To date, only four Acrs, AcrID1, AcrIIIB1, AcrIIIB2, AcrIII-1 and a putative Aca, Aca8, have been identified in archaeal viruses, and all are sparsely distributed (Supplementary Fig. 1)[26,28,33,34]. The *acrID1*, *acrIIIB1*, *acrIIIB2* and *aca8* genes are surrounded by several small, mostly single genes that lack functional annotation (Fig. 1A)[27]. Consequently, the guilt-by-association approach is barely applicable to Acr identification in archaeal viruses due to the small number of known Acrs and the preponderance of uncharacterized proteins (Supplementary Fig. 1).

In an effort to develop an alternative approach to identify Acrs, we made two intriguing observations. First, SIRV2 *aca8* (*gpO1/gp54*) and both known Acr genes, *acrID1* (*gpO3*) and *acrIIIB1* (*gp48*) are expressed immediately post infection of *Sulfolobus islandicus* LAL14/1 (Supplementary Fig. 2A)[35]; second, most of the small hypothetical genes at the SIRV2 genomic termini, including *acrID1* and *acrIIIB1*, share a highly similar regulatory sequence, in a sharp contrast to the very low sequence conservation of the nucleotide sequences of the genes themselves and limited similarity among the amino acid sequences of the encoded proteins (Fig. 1B). The identified consensus regulatory sequence encompasses a core promoter region, with a TATA-box (TTTAWATA) downstream of a TFB recognition element (BRE), a purine rich sequence reported previously as the strongest in binding the archaeal transcription factor TFB (the archaeal homolog of eukaryotic transcription factor TFIIB)[36,37]. Such promoter sequence is also found in *aca8*, albeit within the 5′ end coding sequence of the predicted ORF (83 aa). Multiple sequence alignment of Aca8 homologs showed that the actual translation initiation site is located 84 nucleotides downstream of the currently annotated start codon of *gpO1/gp54* (Supplementary Fig. 2B). Upon re-analysis of the SIRV2 transcriptome, we found that the shorter version of the *gpO1* transcript encoding 55 aa

is the dominant species and only a negligible amount of reads corresponding to the longer transcript was detected in the late stages of viral infection[38]. Thus, the high-level transcription of *aca8*, *acrID1*, *acrIIIB1* and an additional 8 genes among the 13 early genes of SIRV2 appears to be driven from the highly conserved promoter within the consensus regulatory sequence (Fig. 1B). The subsequent repression of the early genes that is observed 1 hour post infection is likely to be due to binding of the predicted wHTH anti-CRISPR associated protein Aca8[39] to the regulatory sequence (Supplementary Fig. 2C) (Fig. 1A), similar to the previously described bacterial Acas[31,40]. Moreover, clustering with *acrs/aca* suggests that the other early transcribed small genes also encode anti-defense proteins. Taken together, these observations prompted us to use the conserved regulatory sequences to predict new anti-defense genes.

### Identification of putative ADGs guided by conserved regulatory sequences

We designed a pipeline for the identification of ADGs in genomes of archaeal viruses (Fig. 1C). Briefly, a 100 bp sequence upstream of the start codon was retrieved from viral single genes, i.e., genes that are not predicted to belong to an operon. Sequences from the same viral genome were aligned using MEME-suite followed by a matrix scan with RSAT to identify genes carrying highly conserved regulatory sequences with BRE and TATA-box characteristic of a strong promoter (see Methods). Only genes that met both of the following criteria were selected as putative ADGs: (1) the presence of a predicted strong promoter; (2) sharing a putative regulatory sequence surrounding the strong promoter in the same viral genome.

We first applied the analysis to *aca8*-carrying rudiviral genomes and predicted 127 ADGs from 17 rudiviruses. Subsequently using the same criteria, we screened the genomes of other members of *Rudiviridae*, i.e, those lacking *aca8*, all members of *Lipothrixviridae*, *Bicaudaviridae* and other unclassified viruses[41–46], leading to the identification of an additional 105 putative ADGs. Among the 232 predicted ADGs, 175 showed no detectable sequence similarity to known archaeal Acrs or Aca8, or any other known proteins (Supplementary Data 1). As anticipated with this strategy, most genes encoding AcrIIIB1 and AcrID1 homologs were identified as ADGs in several viral genomes, clustered with other ADGs, as exemplified by Sulfolobus islandicus filamentous virus (SIFV) and Acidianus rod-shaped virus (Hoswirudivirus ARV2) (Fig. 1D). Notably, we predicted ADGs in viruses, such as Sulfolobales Mexican rudivirus 1 (Mexirudivirus SMRV1), that encode no homologs of known Acrs (Fig. 1D, E and Supplementary Fig. 3). Analysis of gene location in archaeal viruses shows that rudiviruses encode (putative) ADGs at both ends of their linear genomes, whereas lipothrixviral ADGs are predominantly located close to one end of the linear genomes (Fig. 1A, D), and ADGs of bicaudaviruses such as Sulfolobus monocaudaviruses SMV2, SMV3 and SMV4 are located randomly within their circular genomes (Supplementary Fig. 4A).

Similarly, we identified early gene regulatory sequences among 19 putative ADGs (including one *aca8*) within 6 Sulfolobales and Thermofilum MAGs (Metagenome-Assembled Genomes) (Fig. 2, Supplementary Data 1). Among these, a 535 aa ADG (MCI4409744.1) is a fusion of AcrIIIB1 and an unknown domain, with homologs in other MAGs, SMV2 and SIRV isolate V3 (Supplementary Fig. 5). MCI4409744.1 could be one of the largest Acrs, perhaps, endowed with activities against multiple host defense systems.

Archaeal viruses lacking Aca8 homologs have diverse regulatory sequences associated with their putative ADGs (Fig. 1D, E and 2). Although the core promoter, composed of the TATA-box and the BRE element, is highly similar among the consensus sequences, the upstream and downstream regions vary significantly between viruses, which is especially evident within Mexirudivirus SMRV1 and SMV2 motifs (Fig. 1E, Supplementary Fig. 3). The diversity of the regulatory sequences and the absence of an Aca8 homolog together imply that

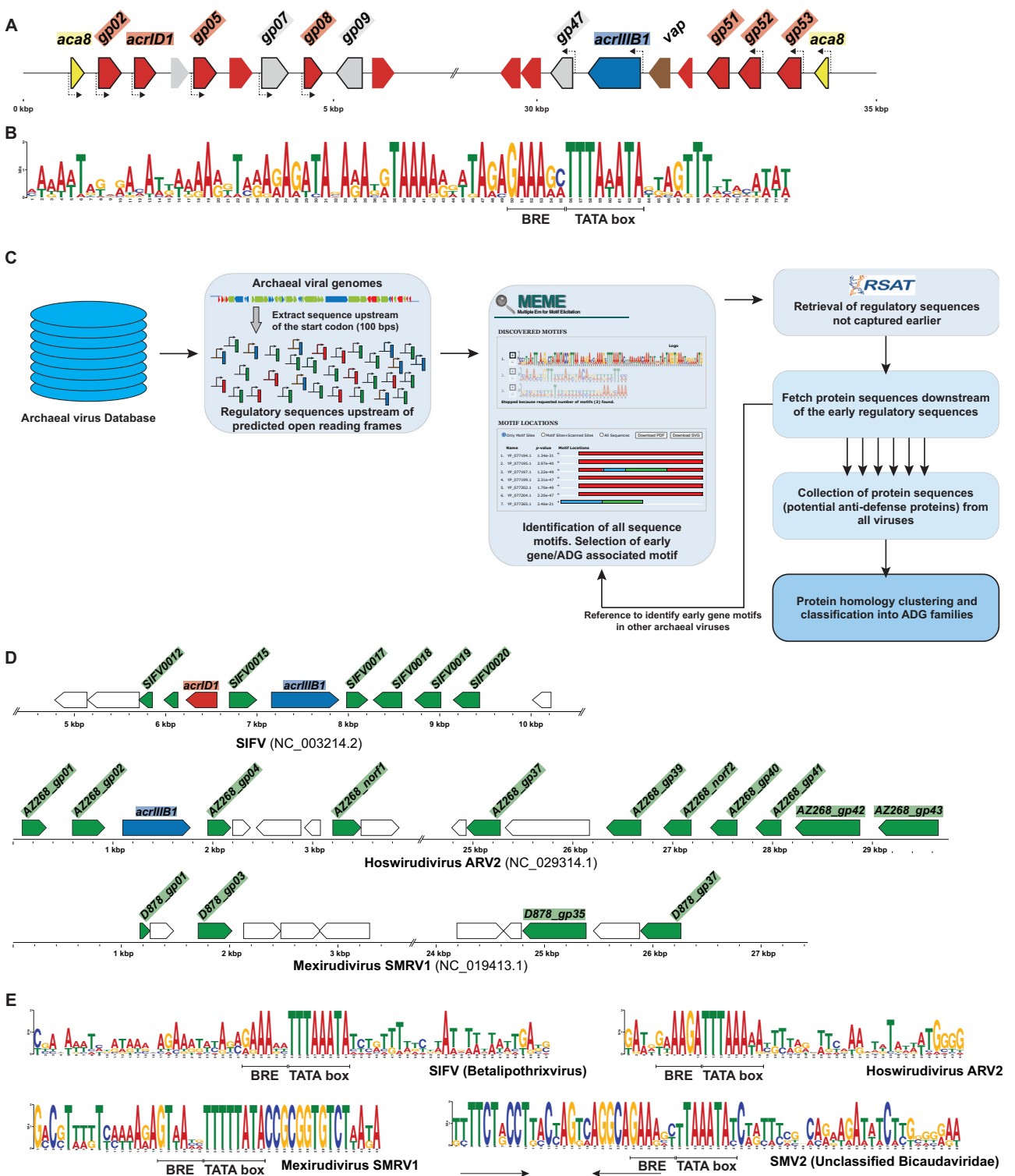

**Fig. 1 | A regulatory sequence encompassing a strong promoter precedes Acr-related genes. A** Illustration of all Icerudivirus SIRV2 genes at both termini including those encoding AcrIIIB1 (blue), AcrID1 homologs (red), Aca8 (yellow), virus-associated pyramid (vap, brown) and hypothetical proteins (gray). Arrow blocks with black lines represent SIRV2 early genes as determined based on their expression pattern (Supplementary Fig. 2A). Dashed arrows indicate the presence of a strong regulatory sequence upstream of the gene. **B** Consensus motif derived from MEME analysis of the regulatory sequences preceding 11 of 13 SIRV2 early genes. **C** Illustration of the general methodology employed here to identify early-genes/anti-defense genes, described in detail in the text. **D** MEME-RSAT prediction of potential ADGs in representative archaeal viral genomes SIFV, Hoswirudivirus ARV2, Mexirudivirus SMRV1 and SMV2. **E** Consensus sequence motifs from selected individual archaeal viruses.

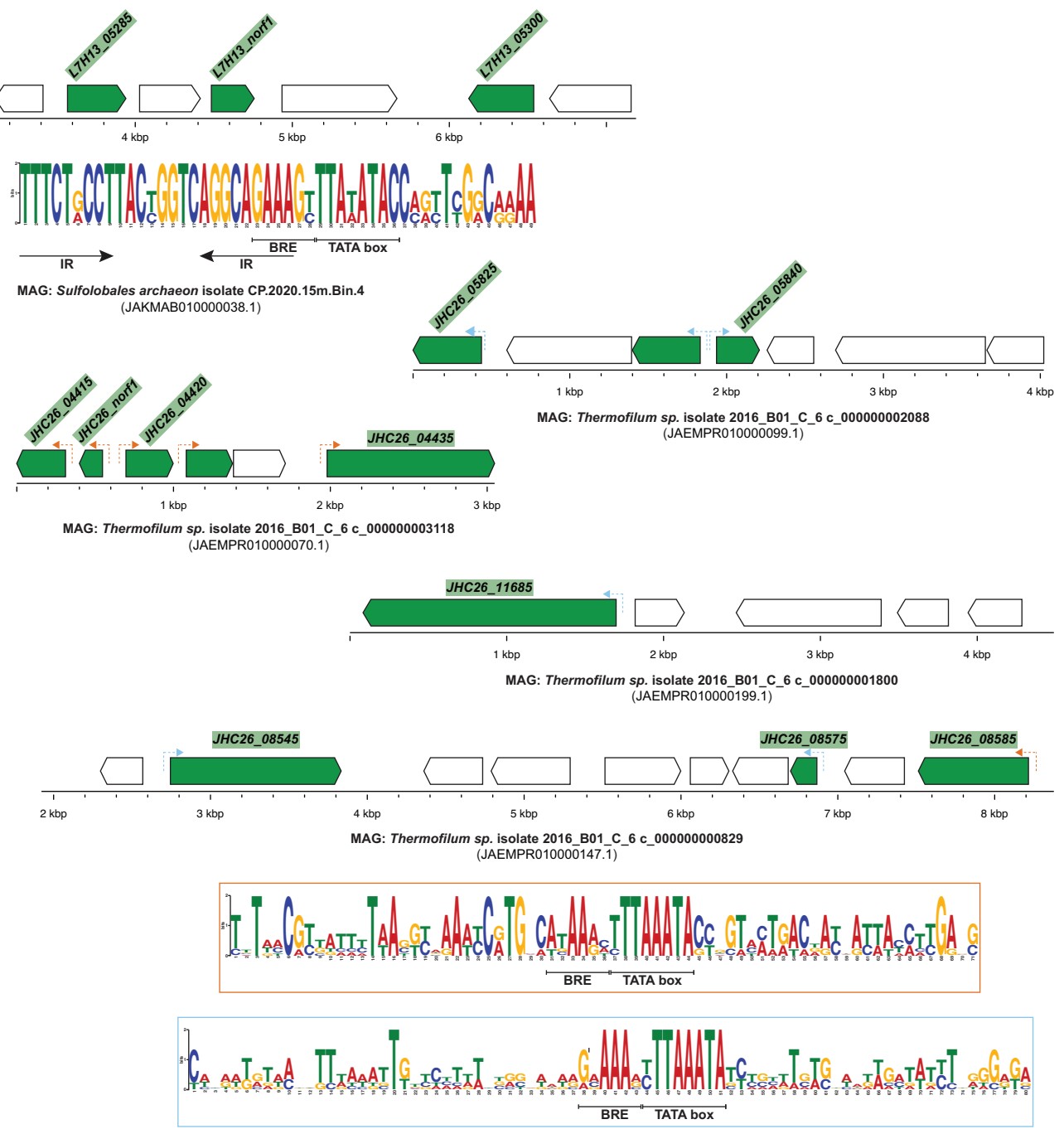

**Fig. 2 | ADGs in metagenome contigs.** Sulfolobales and Thermoprotei contigs, likely to be of viral or plasmid origin, encode the predicted ADGs (shown in green). Two different regulatory sequences were identified from four Thermoprotei contigs and are depicted with dashed arrows upstream of the associated ADGs, in orange and in light blue, respectively. The corresponding consensus regulatory sequence is indicated within a frame of the same color.

these viruses encode different regulatory proteins. As an example, the consensus regulatory sequence identified in SMV2 and the *Sulfolobales archaeon* isolate (MAG: JAKMAB010000038.1) each contains inverted repeats overlapping the BRE element, likely a binding site for a transcriptional repressor (Fig. 1E and Fig. 2). Furthermore, both *SMV2* and the metagenome sequence encode a small protein (SMV2 gp37 and L7H13_norf1, respectively) which is the only predicted ADG shared by these two genomes (Supplementary Data 1). A coiled-coil structure predicted by AlphaFold2, together with the presence of leucine residues spaced at specific intervals point to SMV2 gp37 homologs as leucine zipper proteins, likely binding the inverted repeats to repress transcription of ADGs (Supplementary Fig. 4B).

Some of the ADGs have homologous genes that are not associated with the conserved regulatory sequences, which are therefore not included as ADGs despite amino acid sequence homology with ADG proteins. Next, we calculated motif prevalence, that is, the fraction of homologous genes possessing the regulatory sequence, for all known and newly identified ADGs (Supplementary Data 1). The motif prevalence was 0.75, 0.51, and 0.13 for genes encoding AcrIIIB1, AcrID1 and AcrIII-1 homologs, respectively, and 1 for genes encoding Aca8 homologs, i.e., all *aca8* genes in individual viruses are preceded by the corresponding virus-specific regulatory sequences. While AcrIIIB1 homologs are distributed among many members of two viral families, *Lipothrixviridae* and *Rudiviridae*, mostly as a single copy per genome

(Supplementary Fig. 1), 26 of the 35 identified AcrID1 homologs are encoded in close proximity by only four members of *Rudiviridae*: Iceruduvirus SIRV1 (SIRV1), Iceruduvirus SIRV1 variant XX, SIRV2, and Iceruduvirus SIRV3. This suggests a much higher level of gene duplication and possible functional diversification of AcrID1 which explains the difference in motif prevalence between the AcrIIIB1 and AcrID1 genes. The near lack of association of the ADG-motif with *acrIII-1*, which has one of the lowest motif prevalence among all anti-defense genes, correlates well with the inability of AcrIII (SIRV2 gp37) to function as an Acr in a natural setting[47].

Thus, we developed a new method for predicting ADGs in genomes of archaeal viruses. Using this approach, we identified 251 proteins, including known Acrs, as ADG or ADG-associated genes of which 193 were novel ADGs from 37 archaeal viruses and 6 MAGs. A comparison of the protein sequences encoded by the predicted ADGs to protein sequence databases yielded no homologs for most of them which is not surprising given the generally rapid evolution of anti-defense proteins. Nonetheless, in several of the ADG encoded proteins, helix-turn-helix (HTH) or ribbon-helix-helix (RHH) domains were detected suggesting that their mode of action involves DNA binding (Supplementary Data 1).

## Early gene regulatory motif in Fuselloviruses

The high conservation of the early gene regulatory sequence motivated us to search for similar sequences among temperate archaeal viruses and host genomes. First, we noticed a high sequence identity between the viral early gene promoter and that of the CRISPR-Cas subtype I-A interference gene cluster among Sulfolobales that we studied previously (Supplementary Fig. 6). A RSAT matrix search within the *S. islandicus* LAL14/1 genome identified identical TATA-box and BRE-element in host housekeeping genes encoding the chromatin protein Cren7, S-layer protein SlaA, chaperonin GroEL, a transcriptional regulator, and 8 other host genes. All these genes are known or have the potential to be highly expressed (Supplementary Data 2)[35,48], reinforcing the hypothesis that the ADG motifs (early gene motifs) contain a strong promoter.

*Fuselloviridae* is a family of temperate archaeal viruses with 20 members, SSV1 to SSV22. SSV genes generally form operons and were found to be transcribed into six dominant transcripts and two weak transcripts, referred to as T1-T8[49]. Previously, the BRE sequence of the T6 operon, when placed upstream of TATA-box of the rRNA gene, has been shown to increase rRNA transcription in vitro by about 8 fold, approximately to the level of transcription observed with the T6 promoter[36]. The preferred binding site for TFB contains a consensus sequence of G at −6 and A at −3 positions upstream of the TATA-box start position[36]. We identified a consensus sequence with characteristics of a strong promoter among the regulatory sequences of T6 operons from all of the 20 *SSV*s (Fig. 3A). Together, these features suggest that ADGs are under the control of strong promoters that drive their high expression.

From the 20 SSVs we identified 165 potential ADGs. All four *acrID1* homologs in SSVs that were identified earlier are part of the T6 operons in SSV2, SSV3, and SSV6 (Fig. 3B)[28]. Clustering with known *acr* genes and the high expression level of T6 transcripts during viral infection strongly support the anti-defense function of the novel SSV ADGs.

In total, in all archaeal virus genomes analyzed (57 individual viral genomes and 6 metagenome-assembled genomes), we predicted 354 novel ADGs apart from the previously known Acrs and Acas (Supplemental Table 1). The majority of the ADGs (251) were encoded by single genes, i.e. each with its own promoter and regulatory sequence. Including previously characterized ADGs, all viruses analyzed here carried between 3 and 12 ADGs (Supplementary Fig. 7). Based on amino acid sequence similarity, the ADGs were classified into 116 protein families, ADG.01-ADG.89 encoded as single genes, mADG.01-10 from

MAGs, and ADGSSV.01-ADG.SSV.17 in *SSV* operons. Among the 116 protein families, 42 contained 2 or more members and 74 were singletons. For the 42 non-singleton ADG families, most had an alignment coverage >= 80% with only 8 families falling below this threshold. For most of the families, the amino acid identity (pid) among the within ranged between 20 and 100%, but 11 families included members with a lower percentage identity. To put these findings into context, the AcrID1 family has a minimum coverage of 84.4% and a minimum pid of 7.3%, whereas the AcrIIIB1 family has a minimum coverage of 84.2% and a minimum pid of 20.2%. ADG families with extensive variation in coverage might include members with different domain architectures that could be involved in different anti-defense function.

## A CRISPR-Cas subtype I-A inhibitor in SIFV2 virus

To validate our approach to ADG prediction, we attempted to experimentally identify an inhibitor(s) of subtype I-A CRISPR-Cas system (AcrIA) from the pool of the predicted ADGs (Supplementary Data 1). CRISPR-Cas I-A systems are widespread in archaea, but no AcrIA inhibiting CRISPR interference has been reported. We sought to identify such an Acr and, for this purpose, chose a member of the *Lipothrixviridae* family, Sulfolobus islandicus filamentous virus 2 (SIFV2) (KX467643). SIFV2 is a lytic virus that has not been extensively characterized. It encompasses a cluster of six genes associated with the strong early promoter within the consensus regulatory sequence (see below). Given the almost ubiquitous presence of CRISPR-Cas I-A in *Sulfolobus*, we hypothesize that SIFV2 encodes an AcrIA although in silico analysis revealed only an *acrIIIB1* homolog (Fig. 4A).

To screen the Acr activity of the six SIFV2 ORFs, *gp09, gp10, gp14, gp15, gp16,* and *gp17*, we cloned them individually into the *Sulfolobus - E. coli* shuttle vector pEXA under the control of the weak arabinose promoter, $P_{araS2}$[50]. The plasmids were then transformed individually into the strain *S. islandicus* LAL 14/1 Δ*cas6(I-D)* (hereafter referred to as Δ*cas6(I-D)*) where deletion of the I-D *cas6* gene resulted in inactivation of subtype III-B cmr-γ and subtype I-D whereas subtype I-A and subtype III-B cmr-α remained functional[33,51]. The transformants were infected with four viruses with varied susceptibility to the subtype I-A and subtype III-B cmr-α CRISPR-Cas systems. Among the four viruses, SIRV2M (parental strain) is resistant to both subtype III-B and subtype I-A CRISPR targeting[28,33] and its knockout strain SIRV2MΔ*gp48* (Δ*acrIIIB1*) lost resistance to type III-B targeting[33]. The deletion of a fragment encoding three genes (SIRV2 *gp45-gp47*, adjacent to the AcrIIIB1-coding gene *gp48*) resulted in a virus susceptible to subtype I-A CRISPR targeting which was therefore termed SIRV2MΔ*gp45-47* (Δ*acrIA*)[52]. The fourth strain SIRV2MΔ*gp45-48* (Δ*acrIA*Δ*acrIIIB1*) lacks all four genes and is susceptible to both subtype III-B and subtype I-A CRISPR-Cas targeting.

A preliminary screening employing the weak arabinose promoter showed that among the six tested genes, only *gp15* caused mild host growth retardation upon infection with Δ*acrIA* mutant, indicative of CRISPR-Cas inhibition and ensuing virus propagation (Supplementary Fig. 8). Subsequently, the Acr activity was further demonstrated using a strain expressing *gp15* from a strong arabinose promoter, Δ*cas6(I-D)* p*gp15*$_{araS\text{-SD}}$. In comparison to the empty vector, p*gp15*$_{araS\text{-SD}}$ restored the infectivity of Δ*acrIA* as shown by the significant growth inhibition of the host. However, p*gp15*$_{araS\text{-SD}}$ failed to restore the infectivity of Δ*acrIA*Δ*acrIIIB1* which, in comparison to Δ*acrIA*, lacks *acrIIIB1*. Therefore, the inhibitory activity of *gp15* is restricted to subtype I-A (Fig. 4B). Virus titers from the cultures were quantified to verify that growth retardation was indicative of virus propagation (Fig. 4C). Indeed, p*gp15*$_{araS\text{-SD}}$ enabled the propagation of Δ*acrIA* reaching a level similar to that of the parental virus. To further validate these results, we performed spot assays with serial dilutions of the four viruses on a lawn of host carrying either the empty vector or p*gp15*$_{araS\text{-SD}}$. Except for the parental virus, propagation of the viruses was inhibited in the strain carrying the empty vector (Fig. 4D, left panel). In the strain

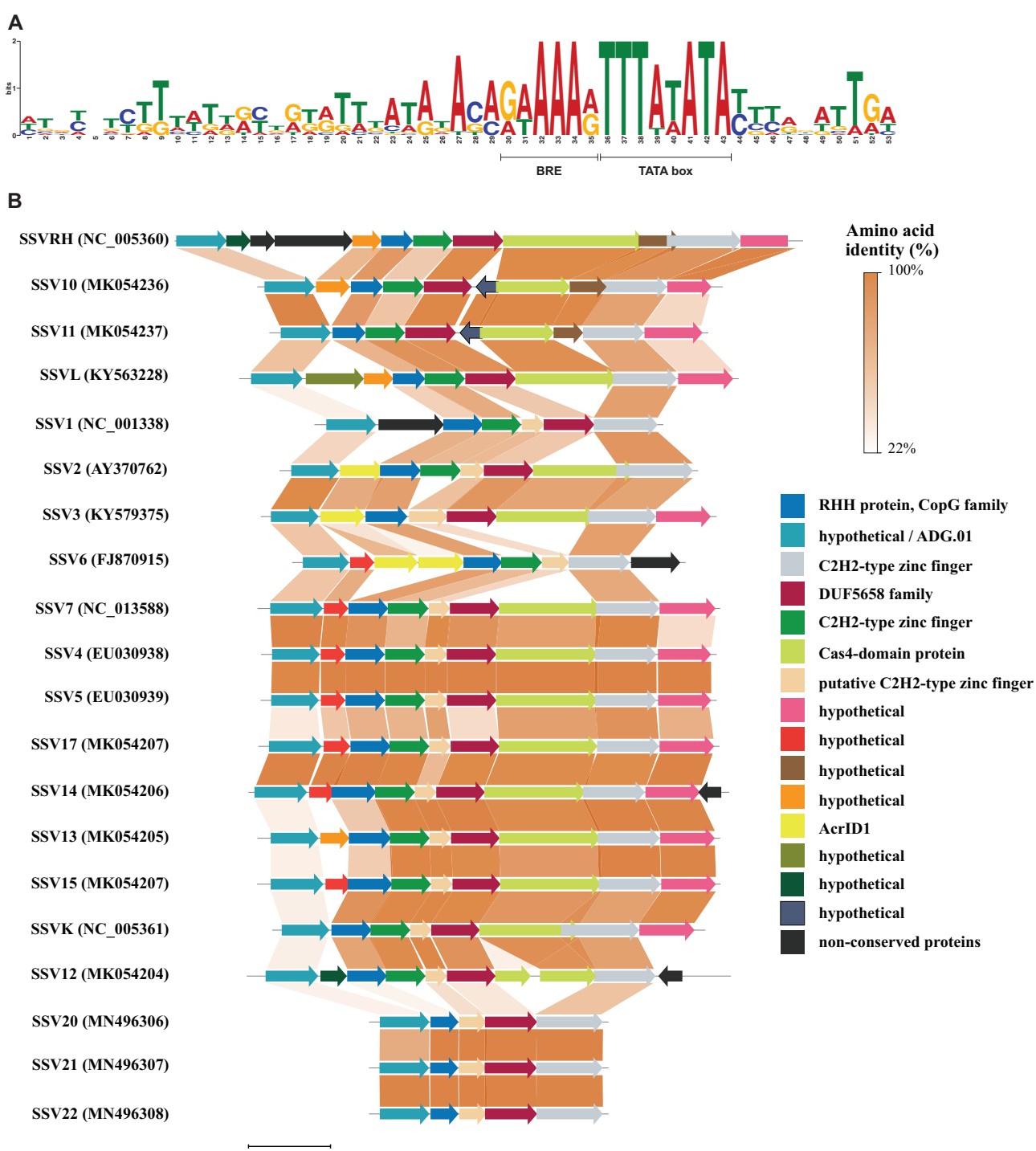

**Fig. 3 | Conservation of an early gene motif in Fuselloviruses. A** Conservation of TATA-box and BRE element within the regulatory sequence of T6 transcript of 20 *Sulfolobus* spindle-shaped viruses. **B** Organization and diversity of the predicted ADGs in the loci encoded by T6 transcript of 20 sequenced SSVs. Homologs are color-coded and their function is expected.

carrying p*gp15*~*araS*-SD (Fig. 4D, right panel), the infectivity of Δ*acrIA* was specifically restored by about four orders of magnitude. Taken together, these results demonstrate that SIFV2 gp15 is an inhibitor of the subtype I-A CRISPR-Cas system.

### A viral ADG inhibits a host toxin-antitoxin system

While most of the 116 ADG families contain exclusively viral genes, ADG.17 (arCOG10132) and ADG.51 (arCOG03737) are not only conserved among viruses infecting diverse hosts, but also are homologous

to some host proteins (Supplementary Data 1, Fig. 5A). Homologs of ADG.17 are encoded in the genomes of Usarudiviruses SIRV7, SIRV8, SIRV9 and SIRV10, and ATSV, SMV1, SMV3, SMV4 and show high similarity to a small protein encoded in several members of the order Sulfolobales (Fig. 5A and Supplementary Fig. 9A)[44,53,54]. The ADG.17 homologs, identified by PSI-BLAST and confirmed by AlphaFold2[55] structural modeling, are encoded in defense islands or integrated plasmids in archaeal genomes (Fig. 5A). A comparison of structural models suggests that these proteins are antitoxins of the Phd (prevents

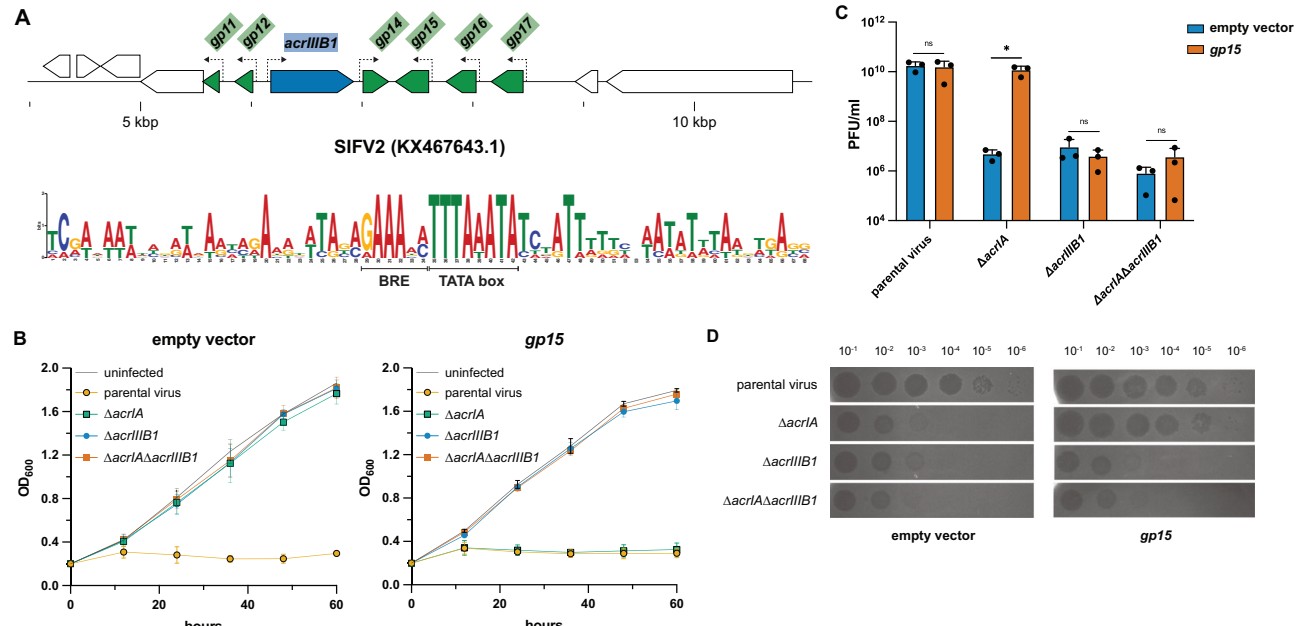

**Fig. 4 | CRISPR-Cas subtype I-A inhibitor in SIFV2. A** Predicted ADGs (top panel) and consensus early gene motif (bottom panel) in SIFV2. **B** Growth curves of *S. islandicus* LAL14/1 Δ*cas6(I-D)* carrying either an empty plasmid (left panel) or a plasmid encoding SIFV2 *gp15* (right panel). Cultures were either not infected (uninfected) or infected with the parental virus (SIRV2M), Δ*acrIA* (SIRV2MΔ*gp45-gp47*), Δ*acrIIIB1* (SIRV2MΔ*gp48*) and Δ*acrIA*Δ*acrIIIB1* (SIRV2MΔ*gp45-gp48*). Data shown are mean of three biological replicates, represented as mean ± SD. Source data are provided as a Source Data file. **C** Virus titer of infected cultures from B at 72 h post viral infection. PFU/ml: plaque forming units per ml of the culture

supernatants. Data shown are mean of three biological replicates, represented as mean value ± SD. A two-tailed unpaired t-test of plaque forming data was used to calculate *P* values; \**P* < 0.05, ns - not significant; *P* = 0.7898, 0.0153, 0.4115, and 0.3373 (from left to right). Source data are provided as a Source Data file. **D** Spot assay of serially diluted virus samples on *S. islandicus* LAL14/1 Δ*cas6(I-D)* carrying either an empty plasmid or a plasmid encoding SIFV2 *gp15*. Dilution fold is indicated on top. The image is representative of five independent experiments. Source data are provided as a Source Data file.

host death) family (Fig. 5B). These genes form convergent gene pairs with genes encoding toxins of the Doc (death on curing) family, known to be kinases inactivating elongation factor Tu[36] (Fig. 5A). Phd and Doc jointly comprise a type II toxin-antitoxin (TA) system which has been well studied in bacteria and phages, but not in archaea[57,58]. As such, ADG.17 appears to be the antitoxin for the Doc toxin in archaea, which remains to be studied experimentally.

To further validate our approach for ADG identification, we characterized one of the viral ADG.17 family proteins, SMV1 gp44. First, we performed detailed analysis of the toxin encoded in the host genome, SiL_0731, and identified a characteristic Fic domain motif HXFX(D/E)(A/G)N(G/K)R, an essential adenylation component[59] (Supplementary Fig. 9B). The structures of the host toxin/antitoxin pair SiL_0730/SiL_0731 and SMV1 gp44/SiL_0731 complexes were modeled using the protein structure and complex prediction program, AlphaFold2[55]. Considering the amino acids conserved between the homologs, we can conclude that inhibition of toxin adenylation by both the cognate antitoxin and the virally encoded homolog occurs through residue D39. This residue interacts with the toxin residues K81 and R82 that are part of the Fic domain motif (Fig. 5B, C and Supplementary Data 3).

Next, we constructed plasmids encoding the *S. islandicus* LAL14/1 Phd-Doc pair SiL_0730/SiL_0731, or either of the two genes, under the transcriptional control of identical arabinose promoters. A similar plasmid with *SiL_0730* (*Phd*) substituted for SMV1 *gp44* was also constructed. As expected, the plasmid encoding the toxin SiL_0731 alone showed a much lower (3 - 4 orders of magnitude) transformation efficiency compared to that of the plasmid encoding both SiL_0730 and SiL_0731. However, the plasmid carrying SiL_0731 and SMV1 gp44 showed a transformation efficiency comparable to that of the plasmid carrying the toxin-antitoxin pair, suggesting an inhibitory effect of the viral small protein on SiL_0730. Subsequently, to identify the nature of

the antitoxin (RNA or protein), we introduced point mutations causing premature translation termination of SMV1 gp44 or SiL_0730. This resulted in a drop in transformation efficiency to levels similar to that of the plasmid encoding the toxin alone, indicating that a protein product of SMV1 gp44 or SiL_0730 is necessary to inhibit the toxicity of SiL_0731 (Fig. 5D).

To investigate a possible direct protein-protein interaction, we performed pull-down assays using histidine tagged SMV1 gp44 or SiL_0730 as the bait, and *E. coli* cell lysate containing heterologously expressed toxin SiL_0731 as prey (Fig. 5E, panels 2 and 3). The toxin copurified with the antitoxin in both cases (Fig. 5E, panels 5–8) confirming direct interaction between Doc and its antitoxin partners.

Homologs of ADG.51 are encoded by Hoswirudivirus ARV3 (ARV3) and Acidianus two-tailed virus 2 (ATV2), and like ADG.17, show high similarity to a host encoded small protein in Sulfolobales, e.g. SiRe_2374 (Supplementary Fig. 10A). SiRe_2374 is encoded in a putative operon together with SiRe_2373, a fusion protein consisting of an N-terminal AAA ATPase domain and a C-terminal PD-DExK superfamily nuclease domain (Supplementary Fig. 10A, Supplementary Data 4). ADG.51 and its homologs are predicted to be wHTH type transcription factors, AlphaFold analysis and a subsequent DALI revealed similarity to transcriptional regulatory proteins of Multiple antibiotic-resistance Repressor (MarR) family (Supplementary Data 4, Supplementary Fig. 10B)[60]. The host proteins homologous to ADG.51 consist of two wHTH domains, whereas the viral homologs contain only the N-terminal wHTH domain, with an additional a-helix. Co-localization of a DNA binding protein and a nuclease as part of a two-gene cassette is typical of type II toxin-antitoxin systems. Hence, we can hypothesize that SiRe_2374/SiRe_2373-like host gene pairs might be a novel type of anti-viral immunity and viruses encode ADG.51 proteins to antagonize the immunity.

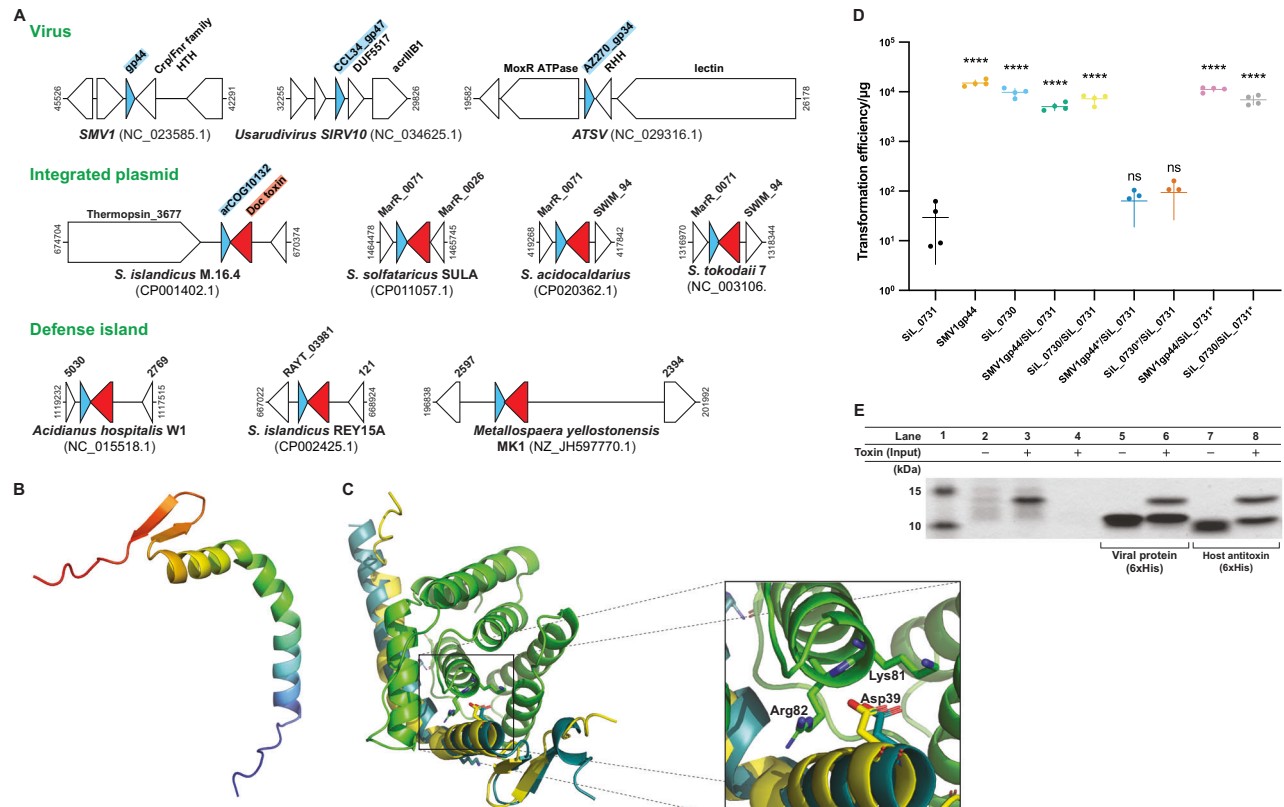

**Fig. 5 | ADG.17 (arCOG10132 domain protein) inhibits host Phd-Doc toxin-antitoxin system. A** Conservation of arCOG10132 domain antitoxin (blue arrows) among archaeal viruses (single gene) and hosts (as part of a two component gene system, Phd-Doc). Doc genes are indicated by red arrows. **B** Alphafold2 predicted structure of the viral antitoxin homolog SMV1 gp44. **C** Structure of SMV1 gp44 (shown in yellow) superimposed on the predicted structure of Phd-Doc (SIL_0730-SIL_0731, shown in deep teal and green respectively). Residue Asp39 conserved among the antitoxin homologs, Lys81 and Arg82 conserved within the toxin homologs. **D** Transformation efficiency in *S. islandicus* of plasmids encoding Doc toxin (*SiL_0731*) or Phd antitoxin (*SiL_0730*), viral homolog of the Phd antitoxin (SMV1 *gp44*) or combinations of toxin and antitoxin including mutants carrying nonsense mutations generating premature stop codons (*). Data shown are mean of

four biological replicates, represented as mean value (center line) ± SD. Logarithm of zero is not possible to plot, hence two values are not shown. A two-tailed unpaired t-test of the transformation data was used to calculate *P* values in relation to the transformation data of SiL_0731; ***$P < 0.001$, ns – not significant; *P* values are <0.0001 (SMV1gp44), <0.0001 (SiL_0730), <0.0001 (SMV1gp44/SiL_0731), <0.0001 (SiL_0730/SiL_0731), 0.1335 (SMV1gp44*/SiL_0731), 0.0671 (SiL_0730*/SiL_0731), <0.0001 (SMV1gp44/SiL_0731*) and <0.0001 (SiL_0730/SiL_0731*). Source data are provided as a Source Data file. **E** Pulldown of the untagged toxin (SIL_0731, lanes 4-8) with histidine tagged antitoxins, SMV1 gp44 (lanes 5-6) and SiL_0730 (lanes 7-8) as bait. Lane 1 – protein ladder. Uninduced and IPTG-induced SiL_0731 cultures are shown in lane 2 and lane 3, respectively. Representative image of two independent experiments, source data provided as a Source Data file.

Taken together, these results indicate that SMV1 gp44 and its homologs in other viruses can act as antitoxins, Phd, to inhibit a host toxin, Doc. Upon infection, the viral Phd antitoxin could efficiently inhibit a possible abortive infection system, Phd-Doc, encoded by their Sulfolobales hosts. Furthermore, homologs of another host protein, co-expressed with AAA+ ATPase/endonuclease domain containing protein and highly conserved among Sulfolobales, is found in ARV3 and ATV2 possibly acting as an antagonist of host anti-viral immunity.

Prompted by these findings we searched for other proteins encoded by viruses of Sulfolobales with significant similarity to host proteins, focusing on small proteins as potential antitoxins (see Methods for details). This search revealed 114 protein families with at least one hit to Sulfolobales genomes including ADG.17 and ADG.51 (Supplementary Data 5). We further examined their genomic context and identified two additional protein families that are consistently encoded next to other small proteins and potentially form toxin-antitoxin gene pairs (Supplementary Fig. 11). One of these families (arCOG07934) is represented in Usarudivirus SIRV8 (YP_009362697.1) and Usarudivirus SIRV9 (YP_009362574.1) and is distantly related to AbrB family of DNA binding proteins (Supplementary Fig. 11A), which function as antitoxins in many known toxin-antitoxin systems[61,62]. Furthermore, in several host genomes, proteins of this family are encoded next to RelE toxins (Supplementary Fig. 11A). Notably,

knockout of arCOG07934 homolog (M164_2845) was found to be lethal in *Sulfolobus islandicus*[63], most likely, due to the activation of the corresponding toxin. Thus, arCOG07934 proteins are confidently predicted to function as antitoxins of RelE toxins. The second example involves proteins of arCOG08091 which are encoded in SIRV1 and SIRV2, and in several Sulfolobales genomes encoded next to proteins from arCOG07288 (Supplementary Fig. 11B). Both proteins are specific for Sulfolobales and do not share any sequence or structural similarity with known proteins. Thus, we hypothesize that arCOG07288-arCOG08091 genes comprise a TA system in which arCOG07288 is the toxin and arCOG08091 is the antitoxin. Alphafold2 modeling of the complex between these proteins suggests that the N-terminal region of the predicted antitoxin forms a beta strand antiparallel to and packing against the C-terminal beta strand of the toxin (Supplementary Fig. 11B). Accordingly, the virus-encoded antitoxin homologs are predicted to inhibit the toxin during infection when the host antitoxin is degraded.

## Discussion

Like most Bacteria, Archaea encode multiple antiviral defense systems, including but not limited to surface resistance, restriction-modification, argonautes and diverse CRISPR-Cas systems. Moreover, many members of Thermoproteota each encode multiple CRISPR-Cas

systems belonging to different types with diverse functionalities. The high diversity of the crenarchaeal defense mechanisms implies a reciprocally complex repertoire of counter-defense genes encoded by archaeal viruses that, in addition to the diversification of the anti-defense proteins, is likely to be subject to elaborate regulation. For example, among the 54 *SIRV2* genes, 25 were classified as accessory genes that are not essential for viral propagation in a CRISPR-Cas deficient laboratory host[64]. It appears likely that most if not all of these accessory genes, including the previously characterized *acrID1* and *acrIIIB1*, are inhibitors of host defense mechanisms. The accessory genes are highly diverse and mostly unrelated to each other, but strikingly, we identified highly conserved regulatory sequences around the promoters of many of these genes, suggesting the existence of early antidefense regulons in archaeal viruses. Notably, the promoter sequence is shared with some highly expressed, house-keeping host genes as well as subtype I-A *cas* genes (Fig. 3A, Supplementary Data 4)[35]. While several of the house-keeping genes preceded by this promoter sequence such as *cren7* are constitutively highly expressed, the strong promoter of CRISPR-Cas I-A interference genes is normally repressed in *Sulfolobus* but derepressed upon viral infection through the release of the repressor Csa3b[51].

Among the 25 SIRV2 accessory genes, 11 genes, located mostly near the genomic termini, are under the control of the typical regulatory sequence with a strong promoter (Fig. 1A, Supplementary Fig. 1C), whereas the rest, as well as the core genes (i.e. genes conserved in all family members), are under the control of distinct, non-early promoters (data not shown). The expression of the early genes including ADGs is energy consuming and therefore must be tightly regulated so as not to interfere with the temporal expression of downstream viral genes[31]. This is essential for viruses to avoid any deterrence to viral yield and to reach the threshold multiplicity that is required to overcome the host defense[29,30,32]. Notably, however, five of the 12 SIRV2 genes encoding AcrID1 homologs lack an early promoter (Fig. 1A). The exact functions of these Acr homologs remain to be explored. Regardless, it appears that viral ADGs diversify not only in the coding region but also in the regulatory sequence, allowing functional fine-tuning at multiple levels.

In particular, the viral ring nuclease gene *acrIII-1* of SIRV1 and SIRV2 is part of an operon encompassing two hypothetical proteins and the holiday junction resolvase[65], which is expressed at the middle infection stage of SIRV2 and lack the early regulatory region identified here. Recent demonstration that AcrIII-1 failed to function as an Acr when expressed from this native promoter in SIRV2 during viral infection of a host carrying type III CRISPR immunity[47] further substantiates the necessity of having an early viral gene expression for CRISPR-Cas inhibition. The functions of homologs of Acrs and other ADGs lacking the early regulatory sequence, however, remain enigmatic.

In this work, we also report the first indications of an archaeal toxin-antitoxin antiviral immune system comprised by a Phd-Doc pair and its virus-encoded inhibitors. In type II TA, the unstable antitoxins are degraded under stresses such as virus infection, and the more stable toxin induces cell dormancy or death which, in the case of infection, protects the rest of the population[66]. Bacteriophages φTE and T4 encode pseudo-ToxI genetic repeats resembling the RNA antitoxin (type III) and the Dmd protein (type II), respectively, to overcome bacterial abortive infection systems by complementing for the degraded host-encoded antitoxins, thereby preventing induction of cellular dormancy or suicide[4,5]. While VapBC, MazEF, and HEPN-NT type antitoxin-toxin systems are highly prevalent in archaea[67,68], our results suggest that the single copy Phd-Doc system is actively involved in communal host defense against viruses. Furthermore, several archaeal viruses have co-opted host Phd antitoxin as an ADG, to inhibit this host abortive infection system. The relatively low motif prevalence score of some of the viral antitoxin homologs

(Supplementary Fig. 4B, Supplementary Data 1) might be explained by the timing of host toxin/antitoxin abortive infection immunity, i.e. post the early virus infection stage. We also identified another potential ADG (ADG.51) encoded in ATV2 and ARV3 and most likely inhibiting a TA. Homologs of this putative ADG were detected in most members of the Sulfolobales where they belong to an operon that also carries a gene encoding a large protein with N-terminal AAA+ ATPase domain and C-terminal PD-DExK endonuclease domain. This operon is likely to encode an abortive infection system resembling the OLD and PARIS systems[69,70], in which the toxin is also a fusion of an ATPase and a nuclease, whereas the ADG.51 homolog is the antitoxin.

Early efforts to predict Acrs using machine learning methods, primarily trained with Acrs from bacteriophages, had limited success in archaeal viruses[9,10]. Recently developed deep learning methods, utilizing basic protein features or pre-trained protein language models, are able to predict thousands of Acrs[71,72]. Among the 116 ADG families identified by our analysis, a total of 79 families were classified as Acrs by AcrNET (62) and DeepAcr (37) including AcrIA SIFV2 gp15 (ADG.56) (Supplementary Data 1). However, it is important to note that not all of these 79 families necessarily include bona fide Acrs, but rather might consist of inhibitors of other defense systems, as exemplified by the putative antitoxin ADG.51 which is classified as an Acr by both methods. Further experiments are needed to explore the functions of all predicted ADGs including the 37 ADG families not annotated as Acrs by DeepAcr and AcrNET which are likely to encode distinct types of anti-defense genes as illustrated by the viral antitoxin ADG.17.

Taken together, the results of this work offer a handle on the hard problem of identifying virus ADGs through the conservation of their regulatory regions including a strong promoter. In retrospect, the conservation of this regulatory sequence makes complete biological sense because, despite the diversity of the ADGs targeting various host defense systems, some of the ADGs have to be expressed at a high level concomitantly at the early stage of infection. Nevertheless, in some cases, the ADG functionality might not depend on its early expression. Prediction of such ADG(s) will not be feasible using the method employed here. In addition, this method requires information on strong promoters which is not always available, and the application of the method is thus limited. Although the presence of false positives such as genes directly involved in the virus life cycle in our ADG list cannot be strictly ruled out, the predicted ADGs are mostly hypothetical, functionally uncharacterized proteins and are not conserved among closely related viral families, making highly unlikely their involvement in essential viral functions. Experimental validation of these predictions, complemented by a comprehensive analysis of the sequences and predicted structures of the putative ADGs, will be a major research program on its own.

## Methods
### Identification of upstream regulatory sequence motifs and putative anti-defense genes
Sequences of 100 bps upstream of the start codon of the individual genes or first gene within the loci were retrieved and analyzed using the MEME suite program (version 5.5.0 or later) to identify conserved motifs (default parameters, minimum motif width set to 30 and maximum motif width set between 75 and 100 bps)[73]. Despite similarities in the TATA-box and BRE among viruses encoding Aca8 (*Iceruidiviruses* (Iceruidivirus SIRV1, Iceruidivirus SIRV1 variant XX, Iceruidivirus SIRV2, and Iceruidivirus SIRV3), *Usarudiviruses* (Usarudivirus SIRV4, Usarudivirus SIRV5, Usarudivirus SIRV8, Usarudivirus SIRV9, Usarudivirus SIRV10, Usarudivirus SIRV11), SIRV6, SIRV7, SIRV isolate V3, SIRV isolate V60 and SIRV isolate V65, *Azorudiviruses* (Azorudivirus SRV) and *Japarudivirus* (Japarudivirus SBRV1), we chose not to combine the flanking sequences to generate a single matrix for the identification of motifs among viruses carrying Aca8 homologs. Instead, we attempted

to identify a specific motif for each individual viral genome because each presumably followed a different evolutionary path and therefore the regulatory sequence is more conserved among ADGs of the same genome. The resulting motif-matrix was used to scan for additional genes carrying the motif within the corresponding viral genome by using the matrix-scan function of the Regulatory Sequence Analysis Tools (RSAT) (default parameters) (Fig. 1C)[74,75]. Viral genes were considered as putative ADGs based on the similarity of their upstream sequence to the virus-specific consensus early regulatory sequence, i.e., a MEME suite p-value lower that e-08 or RSAT e-value lower than e-08.

The genes under possible control of aca8 motifs from different viruses were grouped into clusters of homologous proteins using PSI-BLAST with an e-value of 0.001. The resulting clusters were utilized for approximate identification of Anti-defense related proteins (PSI-BLAST, e-value 0.05) with open-reading frames among archaeal viruses that do not carry a homolog of Aca8 and/or archaeal Acrs. This was done to create reference protein sequences dataset necessary to identify motifs as early depending on the proteins they are associated and to identify, if possible, the approximate location of the ADGs on the viral genome being analyzed.

Using the motif-based strategy we identified individual regulatory sequence consensus with a conserved promoter motif for further 20 viruses including members of *Lipothrixviridae* family (AFV2, AFV3, AFV6, AFV7, AFV8, AFV9, SIFV and SIFV2), *Rudiviridae* (Itarudivirus ARV1, Hoswirudivirus ARV2, Hoswirudivirus ARV3, Mexirudivirus SMRV1), *Ungulaviridae* (Captovirus AFV1), *Bicaudaviridae* (SMV2, SMV3, SMV4, STSV1 and STSV2), *Fuselloviridae* (SseV isolate 1), an unclassified dsDNA virus (SYV1) and six MAGs (JAKMAA010000023.1, JAKMAB010000038.1, JAEMPR010000070.1, JAEMPR010000099.1, JAEMPR010000147.1, and JAEMPR010000199.1). Additionally, we have gathered several ORFs as low probability ADGs which were not included in our total count of novel ADGs. These are ORFs containing TATA-box and BRE with high similarity to the early promoter sequence but listed separately due to their presence among conserved viral core proteins or proximity of the likely Transcription Start Site to the translation start codon. ORFs co-transcribed downstream of the predicted ADGs and those with low prediction values are also included in this list. We predict these ORFs are unlikely to be expressed immediately post infection but might play a role in some cases, depending on the type of defense systems they inhibit. A summary of ADG families and a list of all ADGs in archaeal viruses are provided in Supplementary Data 1. Analysis of the T6 promoter from SSVs was restricted to 20 genomes SSV1 – SSV22. The remaining genomes were omitted due to uncertainty regarding their transcriptional organization.

## Search for putative antitoxin encoded in viral genomes
Sulfolobales viral genomes were downloaded from GenBank and Sulfolobales genomes from RefSeq[76] databases in August 2022. All versus all comparison between viral proteins and proteins from Sulfolobales genomes were performed using PSI-BLAST program with E-value cut-off 1e-10. All proteins were assigned to arCOGs[77] as described previously[78]. Neighborhoods of the identified small proteins (<150 amino acids) were examined manually. If a protein is encoded in a viral context (at least 5 genes with hits to vizarded in the respective neighborhood), it was considered belonging to an integrated virus and was not further considered. Others, if encoded next to other small proteins of the same size, were further analyzed using HHpred[79] to identify remote sequence similarity (if any) and DALI server[80] with respective Alphafold2[55] models to find structural similarity (if any).

## Strains, media and growth conditions
*Sulfolobus islandicus* LAL14/1, *S. islandicus* LAL14/1 Δ*Cas6(I-D)*, *S. islandicus* LAL14/1 Δ*arrays* and their derivatives were grown at 78 °C at 150 rotations per minute (rpm) in SCV medium, as per requirement the medium was supplemented with uracil (20 μg/ml). *Escherichia coli* strains DH5α and Rosetta (DE3) were grown in Lysogeny-Broth (LB) medium at 37 °C, 200 rpm. *E. coli* cultures were supplemented with ampicillin (100 μg/ml), kanamycin (25 μg/ml) and chloramphenicol (10 μg/ml) when necessary.

## Plasmid construction
SIFV2 and SMV1 genomic DNAs were utilized as templates to amplify SIFV2 *gp09, gp10, gp14, gp15, gp16* and *gp17* and SMV1 *gp44*. Purified *S. islandicus* genomic DNA was used as a template for the amplification of *SiL_0730* and *SiL_0731*. The genes were cloned into pEXA2 or pEXA3 plasmids carrying an arabinose promoter without or with a Shine-Dalgarno sequence respectively. Overlap extension PCR was performed using the specified oligonucleotides (Supplementary Table 1), to introduce a nonsense mutation in the coding sequences of *SiL_0730*, *SiL_0731*, and SMV1 *gp44*. All oligonucleotides utilized in this study are listed in Supplementary Table 1.

## Virus titration and spot assay
Supernatants from Δ*Cas6(I-D)*/e.v. and Δ*Cas6(I-D)*/pgp15 cultures infected with SIRV2M and its mutants were sampled 24 h post infection. Virus titration of these samples were performed as described earlier[81]. Briefly, 100 μl of serially diluted supernatants were mixed with 2 ml of Δ*arrays* culture and incubated at 78 °C for 30 minutes. The mixture was then combined with 2 ml of 0.4% Gelzan CM and poured on a pre-heated 0.7% Gelzan CM/SCVU plate. For spot assay, 5 μl of serially diluted pre-titrated virus sample was spotted on 0.7% Gelzan CM/SCV plates and incubated at 78 °C for 2 days.

## Protein purification and pulldown
SMV1 *gp44*, *SiL_0730* and *SiL_0731* were cloned into the prokaryotic expression vector pET30a(+) (Novagen). Oligonucleotides were designed such that upon cloning only SMV1 gp44 and SiL_0730 carried a C-terminal histidine tag while SiL_0731 carried none of the affinity tags. 400 ml cultures of *E. coli* Rosetta(DE3) strains (Novagen) carrying pET30a(+)/*gp44chis* and pET30a(+)/*SiL_0730chis* were grown in LB medium at 37 °C, 200 rpm to $OD_{600} = 0.6$. Cultures were then cooled and protein expression was induced overnight at 15 °C with the addition of 0.5 mM IPTG. Cells were collected by centrifugation at 6300 x g for 10 minutes at room temperature, resuspended in lysis buffer (20 mM HEPES, 300 mM NaCl and 20 mM Imidazole, pH = 7.4) and stored at -80 °C until further processing. Cell pellets, thawed at room temperature, were lysed using a homogenizer (STANSTED model SPCH-10 from homogenizing systems, UK) and centrifuged at $12000 \times g$ for 35 min at 4 °C to remove cell debris. The supernatant was filtered through a 0.45 μm filter prior to purification using a histrap column (HisTrap™ High Performance column, Cytiva). After several washes with lysis buffer, the bound proteins were eluted in a buffer with 250 mM Imidazole and protein purity was analyzed on SDS-PAGE gels. As an additional purification step the proteins were loaded on a Superdex 200 Increase 10/300 gl (Cytiva) equilibrated in a buffer (20 mM Tris, 300 mM NaCl, pH = 8.0). The purified proteins were used in pulldown assays.

The pulldown assay was performed with SiL_0731 as prey. *E. coli* Rosetta(DE3) strain carrying the plasmid pET30a(+)/*SiL_0731* was induced for protein expression as described earlier. Filtered supernatants containing untagged SiL_0731 was split into three and mixed with *SMV1*gp44chis, SiL_0730chis or buffer alone (as control). Two additional controls without SiL_0731 but containing one of either *SMV1*gp44chis or SiL_0730 were also included. The mixtures were then incubated at 65 °C for 30 min, cooled to room temperature and affinity purified. The eluted samples were then analyzed on an SDS-PAGE gel.

## Statistics and reproducibility

The data in all figures are expressed as mean ± standard deviation (SD), statistical significance assessed by two-tailed unpaired t-test. Graph-Pad Prism version 10.0.1 (or later) was used for data analysis. All statistical details pertaining to the experiments, including the number of replicates, statistical significance, and statistical test are indicated in the relevant figure legends.

## Reporting summary

Further information on research design is available in the Nature Portfolio Reporting Summary linked to this article.

## Data availability

The authors declare that the data supporting the findings of this study are available within the paper and its supplementary information files. Sulfolobales viral genomes downloaded from GenBank and Sulfolobus genomes from RefSeq database, are openly accessible from NCBI. The accession IDs of virus genomes and proteins used in the identification of ADGs are listed in Supplementary Data 1. Source data are provided in this paper. Additional information can be obtained from the corresponding authors upon request. Source data are provided in this paper.

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

## Acknowledgements

We thank all members of the Microbial Immunity lab, Department of biology, Copenhagen University for their suggestions. This work was supported by the Novo Nordisk Fonden Postdoctoral Fellowship in Bioscience and Basic Biomedicine Grant (NNF21OC0067491) to Y.B.-C., and by the Danish Council for Independent Research/Natural Sciences [DFF-0135-00402] and Novo Nordisk Foundation/Hallas Møller Ascending Investigator Grant [NNF17OC0031154] to X.P.

## Author contributions

Y.B.-C., L.M.-A., X.P conceived the idea. S. X. performed all the experiments. Y.B.-C., L.M.-A., designed the pipeline for ADG identification and carried out the analyses. S.K. and K.S.M. searched for viral homologs in Sulfolobales host genomes and identified potential antitoxins. Y.B-C, X.P., K.S.M., and E.V.K. supervised the study. All authors contributed to writing of the initial draft. The final manuscript was read and approved by all authors.

## Competing interests

The authors declare no competing interests.
