## [Peer Review File · Nature Communications]

REVIEWER COMMENTS

Reviewer #1 (Remarks to the Author):

Bhooban-Chitty, Xu and colleagues employed an elegant variant of the "guilty by association" strategy to identify anti-defense systems (anti-CRISPRs, Acrs; Anti-CRISPR associated proteins, Acas; anti-defense genes, ADG) in the genomes of viruses infecting archaea. This strategy, commonly used to pinpoint these proteins in Bacteria, is limited by the reliance on already known sequences. The limited information regarding Acrs, Acas, and ADGs encoded by Archaea and their viruses makes their identification challenging, particularly posing a significant constraint on the discovery of novel elements. The authors circumvented these limitations by noting that the few known elements are transcribed during the early phases of viral infection, and further observe that they are under strong transcriptional control often associated with strong promoters. By combining this information, the authors successfully identified numerous genes with putative anti-defense system functions. Additionally, they experimentally validated some of these discovered genes, demonstrating their activity in countering a CRISPR-Cas system and, notably, an inhibitor of an archaeal toxin-antitoxin-based immune system. I share the excitement of these observations, that hint on novel groups of anti-defense genes including an anti toxin-antitoxin archaeal system. The authors consistently support the validity of their hypotheses based on genome sequence and gene expression data, supporting them with exhaustive experimental evidences. I don't identify any severe flaws regarding formulation and validation of the hypotheses (either bioinformatics or experimental validation), however I would like to highlight some minor points that I believe need refinement.

-Despite the detailed content in the manuscript, I found it challenging to read in some instances (long sentences with numerous "commas"). Additionally, typo revisions (e.g., L216 interference).

-The authors frequently refer to clusters of homologous genes/proteins identified in their research. I would recommend adding additional information about the identity of these sequences (e.g., amino acid sequences), for each cluster, to provide an idea of their diversity and evolutionary divergence, along with the criteria for cluster definition.

-I would ask the authors to formulate some (short) hypothesis on how an anti toxin-antitoxin system can evolve (how the virus did acquire it?) given the close dependence of these genes from each other.

-Figure 3. Due to the figure being included in the main text, I would introduce an alignment that clearly displays syntenic regions and protein (sequence) identities. This can be easily accomplished with pyGenomeViz (<https://moshi4.github.io/pyGenomeViz/gui-docs/pgv-gui/>) or similar tools. The same recommendation for Supplementary figure 4, but in this case highlighting the conserved regions.

-Virus names. In nomenclature, the name of the organism that the virus infects is generally written in italics (e.g., *Sulfolobus islandicus* rod-shaped virus 2; L80). In the case of SIRV2, the viral species name should be Icerudivirus SIRV2 (https://ictv.global/taxonomy/taxondetails?taxnode_id=202001547). I recommend that the authors revise all virus names, whether they are referring to the virus species or

the organism that the virus infects.

-Prokaryote clades: same as for virus names. I recommend that the authors adhere to a consistent style when writing the various levels of taxonomy (italicized or not, e.g., Sulfolobales L346 vs L365). Personally, I would use italics only for genus and species, and normal font for the others.

-L389, replace bacteria with Bacteria; archaea with Archaea

Reviewer #2 (Remarks to the Author):

The study by Bhoobalan-Chitty al., suggests a novel approach for prediction of archaeal anti-defence genes. Archaeal anti-CRISPR and anti-defence genes have been poorly characterised due to limited number of available viral sequences and few experimentally validated Acr and Aca proteins. The proposed method relies on the presence of conserved regulatory sequences and strong promoters in early genes of archaeal viruses. Analysis of viral sequences revealed 116 families of potential anti-defence genes. The anti-CRISPR and anti-TA nature of two predicted families were experimentally validated.

The results are interesting and presented in detail. The methods used are adequate, but could be more carefully described.

Authors clearly explained the logic behind their research of anti-viral genes – from early genes of rudiviruses, regulated by Aca8, to other lytic and temperate archaeal viruses. However, by following this logic, the reader is not lead to a summary or “bird’s-eye” view of the results. In that, all the figures in the bioinformatic part show only examples/interesting cases, but the global overview of the new method, predicted ADGs and regulatory motifs is missing. I suggest that the addition of summary figures in the very beginning or just before the experimental part will improve the structure of the article and diversify somewhat repetitive illustrations. I have several suggestions for this analysis:

1) Predicted regulatory sequences are key elements of the article, yet they are not systematically described. Figures provide only partial information on selected motifs in selected viruses (or at least figures make this impression). How many motifs were predicted in total? How many ADGs/families of ADGs are associated with each motif? Are regulatory sequences restricted to one viral family/genera or more widespread?

2) The method for identification of ADGs using regulatory sequences is not sufficiently explained. The procedure described in the main text do not match the Methods section (lines 881-887, blastp search of homologs is not mentioned in the results). I suggest that a schematic of the pipeline could be added as a figure.

3) There should be a greater discussion of limitations of the new method in the manuscript. Is the proposed method specific to anti-defence genes or early viral genes with no defence function could be

predicted as ADGs? How the results are different from machine learning methods for prediction of Acrs?

Minor points:

Line 109-111: According to He et al. 2018, SIRV2 has 12 homologs of AcrID1. In fact, most of the small hypothetical genes at the SIRV2 genomic termini are AcrID1 homologs, including 6 genes with identified regulatory sequence (Figure 1A). The statement that there is "no or very low sequence conservation of the ORFs" is in contradiction with He et al. 2018 and with following analysis of ADG clusters.

Line 112-113: How many different motifs were found preceding SIRV2 early genes by MEME-suite? Are there any potential regulation sequences which did not include TATA-box and BRE?

Line 123-125: How the early genes were defined? In the text, authors mention 13 early genes, 11 early genes in the legend of Figure 1A, and 19 genes on the figure itself. Moreover, in the genome of SIRV2, there are more small genes at the termini which were not included in the Figure 1A (for example gp45 - another homolog of AcrID1). The author should harmonize these data or clearly explain the differences in numbers.

It is hard to follow how many ADGs were identified in total as the number changes through the results:
Line 142-143: "In total, 127 putative ADGs from 17 ruidiviruses were identified, of which 81 (24 families) showed no detectable sequence similarity to known archaeal Acrs or Aca8."

Line 204-206: "Using this approach, we identified 251 proteins, including known Acrs, as ADG or ADG associated genes of which 193 were novel ADGs belonging to 99 families from 37 archaeal viruses and 6 MAGs."

Line 246-249: "In total, apart from known Acrs and Acas, we identified 354 novel ADGs belonging to 116 protein families from 57 archaeal viruses and 6 metagenome-assembled genomes."

Line 150: Missing the word "families" in "these ADGs are hereafter referred to as ADG.01 to ADG.89"?

Line 156-162: Does the distribution of ADGs just correspond to the location of early genes in these viral genomes?

Line 193-194: A logo of this conserved sequence is not shown, only individual examples for SIRV2 and SIRV10 viruses.

Line 193-194: The hypothesis that Aca8 is associated with a specific conserved regulatory sequence should be tested based on co-occurrence of these sequences.

Line 255-256: "no AcrIA inhibiting CRISPR interference has been experimentally characterized" and later mentioning of such inhibitor (Line 273) "and an inhibitor of subtype I-A within the SIRV2 gp45-gp47 gene cluster".

Line 260-261: Are there any I-A spacers in the host strain matching SIFV2?

Line 707: Not all homologs of AcrID1 are shown in red on panel A. The expected position of start codon and number of sequences used in the logo could be added.

Line 707: Some regulatory motifs surrounding the strong promoter are not very convincing (for example SIFV motif). Are these artificially extended around the promoter sequence? Can they be supported by a p-value from MEME-suite?

Line 866-867: What if regulatory sequences are located further than 100 bp from start codon?

Line 869: What is the minimum width of motifs used?

Reviewer #3 (Remarks to the Author):

Discovering new anti-defense systems encoded in archaeal virus genomes is a difficult task due to the low number of already known anti-defense systems, the lack of functional annotation for many viral genes and the weak sequence conservation between viral genes with similar functions. In this work, the authors start from the observation that known anti-defense genes (ADGs) (i) are often early expressed during an infection event and (ii) exhibit a conserved regulatory and promoter sequence. They leverage these observations to develop a method to systematically predict which viral genes could encode ADG. They demonstrate the use of their method to identify candidate anti-CRISPR Cas I-A genes and to validate experimentally the activity of one of the candidates. They also use their approach to identify several viral proteins mimicking antitoxin proteins of toxin-antitoxin systems encoded by their hosts. They validate the antitoxin activity of one such ADG.

The study of antiviral defense systems in bacteria and archaea has exploded in recent years and the knowledge about how viruses counter these defenses is lagging. This work sets up a new systematic approach to identify putative viral genes involved in blocking archaeal defense systems. This approach elegantly exploits a genomic trait which is not very often considered in genomics analyses and goes to the experimental validation on several well-chosen ADG examples. The exploration of the candidate ADGs provided here will be of interest to identify new manners with which viruses tend to escape their host defenses. We want to congratulate the authors on such an exciting work: from a great hypothesis to a convincing execution. We enjoyed reading this and are convinced this will be a very useful study.

Major comments

Given that the authors provide a compelling genomic work, it would be interesting to draw, in a systematic and quantitative manner, the big picture of how ADGs (not all questions need to be answered, would just be nice to have a bit more on that topic). Indeed, given the long list of candidate ADGs provided this seems a great opportunity to describe a bit more about ADGs in a “systematic view” - How are ADGs distributed in viral genomes : how many per genome ? Is there some variance across viral genera/families ? How are they spatially organized ? How much (quantitatively) they tend to cluster together ? Are there hotspots of integration of ADG, if yes, are there often between the same type of core genes across diverse viral genera ?

- What do the predicted ADGs look like : distribution of size of each candidate anti-defense proteins, what domains can we find within them ? How many share similarity with host proteins? It could even be possible to fold the candidates as they are usually very small proteins and describe which structural motifs are found. Also, having the structures of the candidates ADG could help clustering them in a meaningful manner.

- How are they regulated : For the small (typically < 100 a.a.) accessory genes in viral genomes, it could be computed what percentage is under the control of the early promoter/regulatory region identified in this study. Is it the majority (as we would expect given that ADG have to be expressed quickly after DNA injection) ?

Several specific examples are provided at times (for the organization in the genome lines 156-161 + for the structural motifs examples in lines 209-211) but doing it in a more systematic manner would be appreciated given that the work is systematic by essence.

The figures in general could be made more easily understandable:

Some panels could be put in supp mat. For example, in Figure 1D, is it of primary interest for the reader to have the consensus sequence motif for all these viruses ? If the main message is just that it is variable across phages, maybe it doesn't deserve a panel in a main figure.

Some legends in the main figures should be added when needed e.g. in Figure 1A, nothing indicates what the colours of the genes mean in the figure. Reading the legend of Figure 1C to understand what the colours of Figure 1A convey is necessary.

font should be made larger and lines thicker (especially true for Figures 1, 4 and 5).

When giving examples of phage genomes or consensus sequences (e.g. Figure 1D), could the authors provide something to help the reader understand what is the taxonomy of the phage as this is useful information and not easy to grasp.

Sometimes the terminology used in the main text and in the figures is not consistent (e.g. in the hhpred identifiers in Supp Figure 9).

- Supp. Fig. 6 : Is this one biological replicate only ? To show that there is an actual difference in the strain growth carrying the gp15 infected by SIRV2M Δ acrIA vs. parental virus, triplicates and showing error bars would be better because the phenotype is not that strong.

Regarding this experiment, why are viruses Δ acrIA : this at this moment in the analysis has not been proven yet and it is puzzling for the reader as just before is mentioned that "CRISPR-Cas I-A systems are widespread in archaea, but no 256 AcrIA inhibiting CRISPR interference has been experimentally characterized" (line 255). How could the authors know the ADGs encoded by SIRV2M could be acrIA then ?

- Full paragraph on ADG.51 : While, this is elaborated in the discussion, it would have been appreciated to discuss this a bit more during the results paragraph: Feels like a jump to the final hypothesis that SiRe_2374/SiRe73 could be a novel defense system. More information is needed to understand why this hypothesis is formulated: are AAA-ATPase/PD-DEXK nuclease widespread among defense systems ? More specifically in toxin-antitoxin systems ? This is not trivial to everybody. It should also be stated more clearly what is the guess on how this system would work (especially if SiRe2373/ADG.51 are predicted to act as transcriptional regulatory proteins) and how ADG.51 would act on this system.

- Lines 379-386 : To support the hypothesis that arCOG07288-arCOG8091 could act as a toxin-antitoxin system, a quick extra analysis could be performed : to cofold both proteins using AlphaFold as was done for ADG.17. This would provide more clues about whether the two proteins could physically

interact with each other (which is not sufficient to conclude that they form a TA system, but it would give even more weight to the statement).

Minor comments

- Line 114 “reported previously as the strongest in binding the archeal TFB” : Was the reverse search tried? start from the database of TFB recognition elements (BRE) ranked by the strength of binding by TFB and look for other promoters that would indicate early expressed genes/ ?
- Line 124 : How are the 8 putative ADG identified among the 13 early genes ? Because the other 5 genes have already known, non ADG-related, functions ?
- Line 160 : “are located randomly” I would not say “randomly” since they clearly are grouped into two clusters. It would be interesting to highlight which genes are “core” within each viral genus to help identify if the ADG and other “accessory” genes tend to cluster together in the genomes as can be observed in bacteriophages.
- Line 180 : “typical of regulatory sequences involved in transcriptional repression” Need a reference for that (and also it would be good to add a reference for the end of the sentence (Line 185).
- Line 260-262 : It is unclear to me why this SIFV2 encodes precisely an anti-CRISPR Cas I-A gene ? Was it only, because CRISPR Cas I-A are ubiquitous in archaea ? If yes this is applicable to all the archaea viruses. Would this imply that the anti-CRISPR Cas I-A are near ubiquitous in all archaeal viruses ?
- Line 212 : Add “and host genomes” in the title of this section.
- Line 216 : typo “interference” Line 246 : “354 novel ADGs ... from 57 archeal viruses and 6 MAGs” Does this include the 193 ADGs identified before ? Or is it only in the temperate viruses/archeal genomes ?
- Line 246 : “354 novel ADGs ... from 57 archeal viruses and 6 MAGs” Does this include the 193 ADGs identified before ? Or is it only in the temperate viruses/archeal genomes ?
- Line 248 : typo “majority”
- Line 299 : “homologous” is maybe too strong. Maybe similar is more appropriate without a deeper analysis.
- More effort should be put to make the correspondence between the accession ids provided in the raw HHpred results screenshot, and the gene names/accessions provided in the text + reference to Supplementary Figure 9 in the main text, e.g., would be good to specify exactly which panel should be read otherwise it renders the reading tedious.
- Line 337/Figure 5D : Could a point mutation be tried on D39 ? As suggested by the structural analysis.
- Lines 349 and 353 : same comment on the “homologs” term.

We'd like to thank you all for the constructive review of our manuscript. We have addressed all your concerns and changes are highlighted in yellow.

REVIEWER COMMENTS

Reviewer #1 (Remarks to the Author):

Bhooban-Chitty, Xu and colleagues employed an elegant variant of the "guilty by association" strategy to identify anti-defense systems (anti-CRISPRs, Acrs; Anti-CRISPR associated proteins, Acas; anti-defense genes, ADG) in the genomes of viruses infecting archaea. This strategy, commonly used to pinpoint these proteins in Bacteria, is limited by the reliance on already known sequences. The limited information regarding Acrs, Acas, and ADGs encoded by Archaea and their viruses makes their identification challenging, particularly posing a significant constraint on the discovery of novel elements.

The authors circumvented these limitations by noting that the few known elements are transcribed during the early phases of viral infection, and further observe that they are under strong transcriptional control often associated with strong promoters. By combining this information, the authors successfully identified numerous genes with putative anti-defense system functions. Additionally, they experimentally validated some of these discovered genes, demonstrating their activity in countering a CRISPR-Cas system and, notably, an inhibitor of an archaeal toxin-antitoxin-based immune system.

I share the excitement of these observations, that hint on novel groups of anti-defense genes including an anti toxin-antitoxin archaeal system. The authors consistently support the validity of their hypotheses based on genome sequence and gene expression data, supporting them with exhaustive experimental evidences. I don't identify any severe flaws regarding formulation and validation of the hypotheses (either bioinformatics or experimental validation), however I would like to highlight some minor points that I believe need refinement.

-Despite the detailed content in the manuscript, I found it challenging to read in some instances (long sentences with numerous "commas"). Additionally, typo revisions (e.g., L216 interference).

Response: We would like to take this opportunity to express our gratitude to you for the comments. and agree with you on most points. The typos have been corrected and long sentences edited into simpler sentences.

-The authors frequently refer to clusters of homologous genes/proteins identified in their research. I would recommend adding additional information about the identity of these sequences (e.g., amino acid sequences), for each cluster, to provide an idea of their diversity and evolutionary divergence, along with the criteria for cluster definition.

Response: The accession number of all individual proteins identified as ADGs are provided in the Supplementary file 1. An additional tab is now added into the Supplementary file 1 that shows distribution of all ADGs among different families. We have also added a description explaining the criteria and diversity within the families.

-I would ask the authors to formulate some (short) hypothesis on how an anti toxin-antitoxin system can evolve (how the virus did acquire it?) given the close dependence of these genes from each other.

Response: A sentence hypothesizing the likely origin of the viral anti-TA has been added to the existing discussion.

-Figure 3. Due to the figure being included in the main text, I would introduce an alignment that clearly displays syntenic regions and protein (sequence) identities. This can be easily accomplished with pyGenomeViz (<https://moshi4.github.io/pyGenomeViz/gui-docs/pgv-gui/>) or similar tools. The same recommendation for Supplementary figure 4, but in this case highlighting the conserved regions.

Response: The main figure (Figure 3) was modified to show protein identity between different viral homologs. For the Supplementary figure (now Supplementary figure 5) we have changed the color coding to make the figure clearer and included protein identities in comparison to MCI4409744.1.

-Virus names. In nomenclature, the name of the organism that the virus infects is generally written in italics (e.g., *Sulfolobus islandicus* rod-shaped virus 2; L80). In the case of SIRV2, the viral species name should be *Icerodivirus* SIRV2 (https://ictv.global/taxonomy/taxondetails?taxnode_id=202001547). I recommend that the authors revise all virus names, whether they are referring to the virus species or the organism that the virus infects.

Response: We have changed all the virus names as suggested by ICTV.

-Prokaryote clades: same as for virus names. I recommend that the authors adhere to a consistent style when writing the various levels of taxonomy (italicized or not, e.g., *Sulfolobales* L346 vs L365). Personally, I would use italics only for genus and species, and normal font for the others.

Response: The recommended change has been incorporated into the manuscript.

-L389, replace bacteria with *Bacteria*; archaea with *Archaea*

Response: Done.

Reviewer #2 (Remarks to the Author):

The study by Bhoobalan-Chitty al., suggests a novel approach for prediction of

archaeal anti-defence genes. Archaeal anti-CRISPR and anti-defence genes have been poorly characterised due to limited number of available viral sequences and few experimentally validated Acr and Aca proteins. The proposed method relies on the presence of conserved regulatory sequences and strong promoters in early genes of archaeal viruses. Analysis of viral sequences revealed 116 families of potential anti-defence genes. The anti-CRISPR and anti-TA nature of two predicted families were experimentally validated.

The results are interesting and presented in detail. The methods used are adequate, but could be more carefully described.

Authors clearly explained the logic behind their research of anti-viral genes – from early genes of rudiviruses, regulated by Aca8, to other lytic and temperate archaeal viruses. However, by following this logic, the reader is not lead to a summary or “bird’s-eye” view of the results. In that, all the figures in the bioinformatic part show only examples/interesting cases, but the global overview of the new method, predicted ADGs and regulatory motifs is missing. I suggest that the addition of summary figures in the very beginning or just before the experimental part will improve the structure of the article and diversify somewhat repetitive illustrations. I have several suggestions for this analysis:

1) Predicted regulatory sequences are key elements of the article, yet they are not systematically described. Figures provide only partial information on selected motifs in selected viruses (or at least figures make this impression). How many motifs were predicted in total? How many ADGs/families of ADGs are associated with each motif? Are regulatory sequences restricted to one viral family/genera or more widespread?

2) The method for identification of ADGs using regulatory sequences is not sufficiently explained. The procedure described in the main text do not match the Methods section (lines 881-887, blastp search of homologs is not mentioned in the results). I suggest that a schematic of the pipeline could be added as a figure.

3) There should be a greater discussion of limitations of the new method in the manuscript. Is the proposed method specific to anti-defence genes or early viral genes with no defence function could be predicted as ADGs? How the results are different from machine learning methods for prediction of Acrs?

Response: We would like to thank you for the positive comments and suggestions. As suggested by you an illustration of the method has been added to figure 1. The motif identification was not performed on a global scale, sequence motifs were identified from individual viruses. All the motifs identified has now been included in an additional figure (Supplementary figure 3) .

2. This specific description in the methods section (lines 881-887) was only used to identify the possible neighborhood of the anti-defense genes. The sentence has been corrected to clarify this and indicated in the pipeline. Furthermore the entire section

has been rearranged to make the sentencing more clearer and reflect the method as described in the results section.

3. We have included couple of limitations of this method in the discussion. The possibility that some ADGs do not have an early regulatory sequence. Also, we have discussed a scenario where there are no ADGs or if there are other genes that are expressed as early as ADGs, then early viral genes with no defense function would be detected instead of or along with ADGs. Machine learning methods have been predominantly performed in based on previously identified Acrs and rely on features such as small size and co-transcription of closely spaced genes. Few Acrs have been described until now in Archaea and small size is not a characteristic of Archaeal Acrs, hence previous machine learning studies have not identified reliable candidates.

Minor points:

Line 109-111: According to He et al. 2018, SIRV2 has 12 homologs of AcrID1. In fact, most of the small hypothetical genes at the SIRV2 genomic termini are AcrID1 homologs, including 6 genes with identified regulatory sequence (Figure 1A). The statement that there is "no or very low sequence conservation of the ORFs" is in contradiction with He et al. 2018 and with following analysis of ADG clusters.

Response: We meant there was no or very low sequence conservation at nucleotide sequence level, He et al. 2018 picked up the 12 AcrID1 homologs based on aa sequence homology. We have now clarified the confusion.

Line 112-113: How many different motifs were found preceding SIRV2 early genes by MEME-suite? Are there any potential regulation sequences which did not include TATA-box and BRE?

Response: MEME-suite identified 11 early gene motifs in SIRV2 (including 2 duplicates within the ITR at either ends). Through MEME-suite/RSAT motif analysis we did not identify any regulatory motif lacking the TATA-box and BRE.

Line 123-125: How the early genes were defined? In the text, authors mention 13 early genes, 11 early genes in the legend of Figure 1A, and 19 genes on the figure itself. Moreover, in the genome of SIRV2, there are more small genes at the termini which were not included in the Figure 1A (for example gp45 - another homolog of AcrID1). The author should harmonize these data or clearly explain the differences in numbers.

Response: We characterized SIRV2 early genes based on their expression pattern (from SIRV2 infection transcriptomics performed by Quax et al 2013). We identified 13 genes that peaked at 1 hpi and were subsequently repressed along the virus life cycle (supplementary figure 2A). We only labeled the 11 early genes that carry the early

regulatory motifs, and we agree that it's more appropriate to label all the 13 genes which is now done in the revised figure. Gp45 and gp46 are both homologs of AcrID1 but are not early genes. They are now included in the revised figure as AcrID1 homologs. Their expression pattern is not similar to that of the early genes and hence unlikely to be early gene which correlates well with the lack of the strong regulatory sequence.

It is hard to follow how many ADGs were identified in total as the number changes through the results:

Line 142-143: "In total, 127 putative ADGs from 17 rudiviruses were identified, of which 81 (24 families) showed no detectable sequence similarity to known archaeal Acrs or Aca8."

Line 204-206: "Using this approach, we identified 251 proteins, including known Acrs, as ADG or ADG associated genes of which 193 were novel ADGs belonging to 99 families from 37 archaeal viruses and 6 MAGs."

Line 246-249: "In total, apart from known Acrs and Acas, we identified 354 novel ADGs belonging to 116 protein families from 57 archaeal viruses and 6 metagenome-assembled genomes."

Response: We agree with you, and we have simplified the description.

Line 150: Missing the word "families" in "these ADGs are hereafter referred to as ADG.01 to ADG.89"?

Response: Done.

Line 156-162: Does the distribution of ADGs just correspond to the location of early genes in these viral genomes?

Response: Transcriptomic data are available only for several archaeal viruses, hence we have limited information for this question. Nevertheless, the distribution of SIRV2 ADGs does correspond to the location of early genes but not all genes in the neighborhood carry the early regulatory sequences, as mentioned earlier by you (there are several small genes present in the neighborhood which are not ADGs).

Line 193-194: A logo of this conserved sequence is not shown, only individual examples for SIRV2 and SIRV10 viruses.

Line 193-194: The hypothesis that Aca8 is associated with a specific conserved regulatory sequence should be tested based on co-occurrence of these sequences.

Response: Apologies for the mistake/confusion, the Aca8 homologs from different viruses all have the corresponding virus specific motif (identifying them as ADG in the individual viruses they are encoded on the individual viruses), unlike AcrID1 and AcrIIB1 which are not always identified as ADG. The sentence is rephrased to provide a better explanation.

Line 255-256: "no AcrIA inhibiting CRISPR interference has been experimentally characterized" and later mentioning of such inhibitor (Line 273) "and an inhibitor of subtype I-A within the SIRV2 gp45-gp47 gene cluster".

Response: We apologize for the lack of clarity, which is now included in the revised version as "The deletion of a fragment encoding three genes (*SIRV2 gp45-gp47*, adjacent to the AcrIIIB1-coding gene *gp48*) resulted in a virus susceptible to subtype I-A CRISPR targeting which was therefore termed *SIRV2MΔgp45-47 (ΔacrIA)* (Bhoobalan-Chitty et al., unpublished). The fourth strain *SIRV2MΔgp45-48 (ΔacrIAΔacrIIIB1)* lacks all four genes and is susceptible to both subtype III-B and subtype I-A CRISPR-Cas targeting"

Line 260-261: Are there any I-A spacers in the host strain matching SIFV2?

Response: Yes, there are two spacers against SIFV2 in *S. islandicus* LAL14/1, one spacer in the I-A repeat containing array (two mismatches) and one in the I-D repeat containing array (five mismatches). It should be noted that the virus SIFV2 was never used for infection, potential ADG candidates were expressed on a plasmid and infected with SIRV2M mutant lacking Acrs and susceptible to I-A targeting.

Line 707: Not all homologs of AcrID1 are shown in red on panel A. The expected position of start codon and number of sequences used in the logo could be added.

Response: All homologs of AcrID1 are now shown in panel A of Figure 1. The likely start codon position is variable among individual genes and viruses. For SIRV2, the approximate start position is shown in supplementary figure 2. Upstream sequences from all SIRV2 genes were used to identify the regulatory sequences (logo shown in supplementary figure 2).

Line 707: Some regulatory motifs surrounding the strong promoter are not very convincing (for example SIFV motif). Are these artificially extended around the promoter sequence? Can they be supported by a p-value from MEME-suite?

Response: All motifs were generated by MEME suite based on the 100 bps upstream sequences, we did not extend any promoter sequences. For all viruses, a cutoff of $1e-08$ was set for the p-value. As an example, the meme-suite p-values are shown in supplementary figure 1C for the early SIRV2 genes. Below you can also find the RSAT values (pval) for identification of the ADG in SIFV, all SIFV genes listed in the supplementary file are identified with a p-value above $1e-10$.

ft_name	strand	start	end	sequence	weight	Pval	In_Pval	sig	rank
START_END	D	-48980	-1	.	0	0	0	0	
CBAAWMTDRWAAAHAGAARTWYABAGAAAAATTTAAATATCTRTTMDCKMNRWTWMTKABV	R	-31460	-31394	CGAAAAATCGATAAAAAAGAGTAGAGAAATCTTTAAATATCTATGTCTAATGTATTAATGACC	43.9	4.2e-24	-53.834	23.388	1
CBAAWMTDRWAAAHAGAARTWYABAGAAAAATTTAAATATCTRTTMDCKMNRWTWMTKABV	R	-32312	-32246	CGAAAAATCGATAAGGAGAAAGACACAGAAAAATTTAAATATCTCTTATCAATATACAATCGACC	41.5	5.3e-22	-48.997	21.279	2
CBAAWMTDRWAAAHAGAARTWYABAGAAAAATTTAAATATCTRTTMDCKMNRWTWMTKABV	D	-33807	-33741	CGAGAAATCGAGCAAAAGAGTATATCGAAAAATTTAAATCCGGTTTATGTATATCTTGTAGT	41.0	1.3e-21	-48.082	20.882	3
CBAAWMTDRWAAAHAGAARTWYABAGAAAAATTTAAATATCTRTTMDCKMNRWTWMTKABV	D	-34275	-34209	TCTTAGATATAAGAGCAGAAAGTTCATCGAAAAATTTAAATATATGTTTTACGTATGTAATGATC	36.7	1.3e-18	-41.182	17.885	4
CBAAWMTDRWAAAHAGAARTWYABAGAAAAATTTAAATATCTRTTMDCKMNRWTWMTKABV	D	-32981	-32915	CGAGAAAAATGTTCTAGAGATTTGTCGAAAAATTTAAATATTTATTTTGGTATTTTTAAATGATA	35.3	9.2e-18	-39.223	17.835	5
CBAAWMTDRWAAAHAGAARTWYABAGAAAAATTTAAATATCTRTTMDCKMNRWTWMTKABV	R	-31880	-31814	CCATTTTTTCATGAATCAAAAAATAGAGAAATCTTTAAATATCTATTTTCTAATGATAATGAGGG	34.0	5.2e-17	-37.494	16.283	6
CBAAWMTDRWAAAHAGAARTWYABAGAAAAATTTAAATATCTRTTMDCKMNRWTWMTKABV	R	-35840	-34974	CATTACTGAAGCAACTAACGCATACAGAAAAATTTAAATCAATGTGCTACAATTTATGATAGA	29.4	1.2e-14	-32.067	13.926	7
CBAAWMTDRWAAAHAGAARTWYABAGAAAAATTTAAATATCTRTTMDCKMNRWTWMTKABV	R	-34330	-34264	TTATATCTAAGATCTAGAAATCAAGAAAAAGATTTAAATAAGAGTAGAAGAACATTATGTTAGG	23.7	3.3e-12	-26.424	11.476	8
CBAAWMTDRWAAAHAGAARTWYABAGAAAAATTTAAATATCTRTTMDCKMNRWTWMTKABV	R	-34760	-34694	CTCCCTTCTTAAACTTCTTTTCGAAAAAGTTTTAAATCCGCTAATTTCTAAATATATTTAGG	21.6	2.1e-11	-24.568	10.670	9
CBAAWMTDRWAAAHAGAARTWYABAGAAAAATTTAAATATCTRTTMDCKMNRWTWMTKABV	D	-32215	-32149	TCAGAAATCAGAGAAATAGAAATAAAAAAGGAGAAAAATTTTTTCATAATTAGAATATATA	4.6	2.3e-06	-12.980	5.637	10
CBAAWMTDRWAAAHAGAARTWYABAGAAAAATTTAAATATCTRTTMDCKMNRWTWMTKABV	D	-13879	-13813	AATGAAGTTACACTAGTACATTTATAGCAATTTAAGAAGTAAAGTAGTACACATTAGCAGAGA	3.6	3.9e-06	-12.449	5.487	11
CBAAWMTDRWAAAHAGAARTWYABAGAAAAATTTAAATATCTRTTMDCKMNRWTWMTKABV	R	-18737	-18671	CTTGAATTCGTGCAATCTTACAGGGAAAGATATAATATATAAATTTGCAAAAGATGGGAGA	3.5	4.2e-06	-12.391	5.382	12
CBAAWMTDRWAAAHAGAARTWYABAGAAAAATTTAAATATCTRTTMDCKMNRWTWMTKABV	D	-18993	-18927	GGGGTAAATATGATGTTAGGGACATAAGAAATTTAAAGAAATTTAGAAAGAAATTTAGTAATTC	2.5	7.0e-06	-11.877	5.158	13
CBAAWMTDRWAAAHAGAARTWYABAGAAAAATTTAAATATCTRTTMDCKMNRWTWMTKABV	D	-27165	-27099	CGTGATTTAATTTATATATTTATTGAGAAAGATTTATAGTCTTTTGTATGCTCATATGTC	1.8	9.9e-06	-11.524	5.005	14

Line 866-867: What if regulatory sequences are located further than 100 bp from start codon?

Response: We also thought about this. Therefore, a RSAT analysis on the viral genome, using the motif specific matrix file generated from MEME suite, was always performed to identify any regulatory sequences that were missed in the initial stage. The final data shows that on the whole most regulatory sequences were present within 100 bps upstream of the start codon.

Line 869: What is the minimum width of motifs used?

Response: In general, we restricted the motif width to a minimum of 30 bps in most cases except in a few cases where only the TATA-box and BRE element were the only consensus.

Reviewer #3 (Remarks to the Author):

Discovering new anti-defense systems encoded in archaeal virus genomes is a difficult task due to the low number of already known anti-defense systems, the lack of functional annotation for many viral genes and the weak sequence conservation between viral genes with similar functions. In this work, the authors start from the observation that known anti-defense genes (ADGs) (i) are often early expressed during an infection event and (ii) exhibit a conserved regulatory and promoter sequence. They leverage these observations to develop a method to systematically predict which viral genes could encode ADG. They demonstrate the use of their method to identify candidate anti-CRISPR Cas I-A genes and to validate experimentally the activity of one of the candidates. They also use their approach to identify several viral proteins mimicking antitoxin proteins of toxin-antitoxin systems encoded by their hosts. They validate the antitoxin activity of one such ADG. The study of antiviral defense systems in bacteria and archaea has exploded in recent years and the knowledge about how viruses counter these defenses is lagging. This work sets up a new systematic approach to identify putative viral genes involved in blocking archaeal defense systems. This approach elegantly exploits a genomic trait which is not very often considered in genomics analyses and goes to the experimental validation on several well-chosen ADG examples. The exploration

of the candidate ADGs provided here will be of interest to identify new manners with which viruses tend to escape their host defenses. We want to congratulate the authors on such an exciting work: from a great hypothesis to a convincing execution. We enjoyed reading this and are convinced this will be a very useful study.

Response: We would like to take this opportunity to thank you for the kind comments about our manuscript.

Major comments

Given that the authors provide a compelling genomic work, it would be interesting to draw, in a systematic and quantitative manner, the big picture of how ADGs (not all questions need to be answered, would just be nice to have a bit more on that topic). Indeed, given the long list of candidate ADGs provided this seems a great opportunity to describe a bit more about ADGs in a “systematic view”

- How are ADGs distributed in viral genomes : how many per genome ? Is there some variance across viral genera/families ? How are they spatially organized ? How much (quantitatively) they tend to cluster together ? Are there hotspots of integration of ADG, if yes, are there often between the same type of core genes across diverse viral genera ?

- What do the predicted ADGs look like : distribution of size of each candidate anti-defense proteins, what domains can we find within them ? How many share similarity with host proteins? It could even be possible to fold the candidates as they are usually very small proteins and describe which structural motifs are found. Also, having the structures of the candidates ADG could help clustering them in a meaningful manner.

- How are they regulated : For the small (typically < 100 a.a.) accessory genes in viral genomes, it could be computed what percentage is under the control of the early promoter/regulatory region identified in this study. Is it the majority (as we would expect given that ADG have to be expressed quickly after DNA injection) ? Several specific examples are provided at times (for the organization in the genome lines 156-161 + for the structural motifs examples in lines 209-211) but doing it in a more systematic manner would be appreciated given that the work is systematic by essence.

Response: In general, we agree with you that a systematic view would be informative and useful, but some of these require extensive bioinformatic analysis and an extended description, something we collectively believe would be more suitable as part of a future study. Nevertheless, we have now answered a few of your questions, we have also included a supplementary figure to showing the distribution of ADGs across individual genomes (now supplementary figure 7), there are hotspots of ADGs, for example the termini for rudiviruses.

A systematic analysis of ADG and other small proteins for their similarity to host proteins has already been done (line 365 – line 386). All the ADGs described here are expected to be under the control of early regulatory sequences (a requirement for our analysis). Archaeal viruses carry several small genes (for example SIRV2

vap- virus associated pyramid protein) which neighbor ADGs and not be expressed early upon injection. Our intent was only to provide a general idea in these sentences. We agree that a detailed analysis would be interesting, as mentioned earlier perhaps as part of a more dedicated study.

The figures in general could be made more easily understandable:

Some panels could be put in supp mat. For example, in Figure 1D, is it of primary interest for the reader to have the consensus sequence motif for all these viruses ? If the main message is just that it is variable across phages, maybe it doesn't deserve a panel in a main figure.

Some legends in the main figures should be added when needed e.g. in Figure 1A, nothing indicates what the colours of the genes mean in the figure. Reading the legend of Figure 1C to understand what the colours of Figure 1A convey is necessary.

font should be made larger and lines thicker (especially true for Figures 1, 4 and 5). When giving examples of phage genomes or consensus sequences (e.g. Figure 1D), could the authors provide something to help the reader understand what is the taxonomy of the phage as this is useful information and not easy to grasp.

Response: We agree with you, a supplemental PDF with all the sequence motifs along with the taxonomic details has been added and only a few representative motifs have been retained in the main figure. The legends have been corrected and figures improved visually.

Sometimes the terminology used in the main text and in the figures is not consistent (e.g. in the hhpred identifiers in Supp Figure 9).

Response: The figures have been modified to be consistent with the text. Also, additional designations were included on figures to make it clear which protein IDs correspond to alignments and HHpred outputs.

- Supp. Fig. 6 : Is this one biological replicate only ? To show that there is an actual difference in the strain growth carrying the gp15 infected by SIRV2M Δ acrIA vs. parental virus, triplicates and showing error bars would be better because the phenotype is not that strong.

Response: The experiments were repeated under different experimental conditions. We have modified the figure (now Supplementary figure 7) to include an average of two biological replicates and corresponding error bars. Although the phenotype is not that strong it is consistent across the supplementary figure and the main figure (where gp15 shows a strong Acr activity).

Regarding this experiment, why are viruses Δ acrIA : this at this moment in the analysis has not been proven yet and it is puzzling for the reader as just before is mentioned that "CRISPR-Cas I-A systems are widespread in archaea, but no 256 AcrIA inhibiting CRISPR interference has been experimentally characterized" (line

255). How could the authors know the ADGs encoded by SIRV2M could be acrla then ?

Response: Apologies for not providing a clearer explanation. Lack of SIRV2 gp45-gp47 makes the virus non-infectious in *S. islandicus* LAL14/1 Δ cas6(I-D) (encoding functional type I-A and type III-B) despite the presence of AcrIIIB1. There is further conclusive evidence that one of the three genes is an anti-CRISPR inhibiting the subtype I-A, which is part of a manuscript in preparation. We have clarified this in the revised version.

- Full paragraph on ADG.51 : While, this is elaborated in the discussion, it would have been appreciated to discuss this a bit more during the results paragraph: Feels like a jump to the final hypothesis that SiRe_2374/SiRe73 could be a novel defense system. More information is needed to understand why this hypothesis is formulated: are AAA-ATPase/PD-DExK nuclease widespread among defense systems ? More specifically in toxin-antitoxin systems ? This is not trivial to everybody. It should also be stated more clearly what is the guess on how this system would work (especially if SiRe2373/ADG.51 are predicted to act as transcriptional regulatory proteins) and how ADG.51 would act on this system.

Response: We have added sentences to describe the transition between functional prediction of individual proteins to the hypothesis that this system might function as an host encoded anti-viral defense system.

- Lines 379-386 : To support the hypothesis that arCOG07288-arCOG8091 could act as a toxin-antitoxin system, a quick extra analysis could be performed : to cofold both proteins using AlphaFold as as was done for ADG.17. This would provide more clues about whether the two proteins could physically interact with each other (which is not sufficient to conclude that they form a TA system, but it would give even more weight to the statement).

Response: We made a model for the complex of arCOG07288-arCOG8091 which indeed shows how two proteins might interact. We now show the model of this complex in the updated Supplementary Figure 11 and included the following sentence in the main text: "AlphaFold2 modelling of the complex between these proteins suggests that the N-terminal region of the predicted antitoxin forms a beta strand antiparallel to and packing against the C-terminal beta strand of the toxin (Supplementary Figure 11B)."

Minor comments

- Line 114 "reported previously as the strongest in binding the archeal TFB" : Was the reverse search tried? start from the database of TFB recognition elements (BRE) ranked by the strength of binding by TFB and look for other promoters that would indicate early expressed genes/ ?

Response: We were not able to find any data that systematically rank the strength of different promoters and their corresponding binding strengths to TFB in Archaea.

- Line 124 : How are the 8 putative ADG identified among the 13 early genes ? Because the other 5 genes have already known, non ADG-related, functions ?

Response: The 8 putative ADGs are in addition to 3 known Acr/Aca, so in total 11 out of 13. They were identified as ADGs due to the consensus in their regulatory sequence necessary to maintain their high expression level, a characteristic of anti-host defense genes.

- Line 160 : “are located randomly” I would not say “randomly” since they clearly are grouped into two clusters. It would be interesting to highlight which genes are “core” within each viral genus to help identify if the ADG and other “accessory” genes tend to cluster together in the genomes as can be observed in bacteriophages.

Response: The core genes are now highlighted. In comparison to other archaeal viruses and the distribution of core genes, in specific those with circular genomes (like Fuselloviridae) there is some randomness.

- Line 180 : “typical of regulatory sequences involved in transcriptional repression” Need a reference for that (and also it would be good to add a reference for the end of the sentence (Line 185).

Response: While there are several examples of transcriptional regulators binding inverted repeats, pinpointing specific review articles for reference proves challenging. As an alternative, we have adjusted the sentence to convey a more speculative tone.

- Line 260-262 : It is unclear to me why this SIFV2 encodes precisely an anti-CRISPR Cas I-A gene ? Was it only, because CRISPR Cas I-A are ubiquitous in archaea ? If yes this is applicable to all the archaea viruses. Would this imply that the anti-CRISPR Cas I-A are near ubiquitous in all archaeal viruses ?

Response: The chances of a virus encoding any Acr/Adg is based on the host it infects. CRISPR-Cas subtype I-A is widespread in archaea hence the likely hood that SIFV2 encodes an anti-CRISPR of subtype I-A is high. SIFV2 also encodes a homolog of AcrIIIB1, subtype III-B is also widely distributed in Sulfolobus. Yes, we agree that AcrIA is probably ubiquitous in all archaeal viruses.

- Line 212 : Add “and host genomes” in the title of this section.

Response: Only the TATA-box and BRE is present in the host genomes. The consensus regulatory sequence (including the TATA/BRE) motif is prevalent in the temperate virus and not the host genomes.

- Line 216 : typo “interference”

Response: typo corrected, thank you.

- Line 246 : “354 novel ADGs ... from 57 archeal viruses and 6 MAGs” Does this include the 193 ADGs identified before ? Or is it only in the temperate viruses/archeal genomes ?

Response: The 354 is the total count including both the 193 identified before and those from temperate archaeal viruses. Changes have been made to clarify.

- Line 248 : typo “majority”

Response: Corrected.

- Line 299 : “homologous” is maybe too strong. Maybe similar is more appropriate without a deeper analysis.

Response: we agree, and have changed to similar.

- More effort should be put to make the correspondence between the accession ids provided in the raw HHpred results screenshot, and the gene names/accessions provided in the text + reference to Supplementary Figure 9 in the main text, e.g., would be good to specify exactly which panel should be read otherwise it renders the reading tedious.

Response: The text has been modified to indicate specific Supplementary Figure 9 panels and protein accessions are now consistent between the figure and the text. See also response above.

- Line 337/Figure 5D : Could a point mutation be tried on D39 ? As suggested by the structural analysis.

Response: The high conservation of this residue and previous reports in bacteria on the role of negatively charged residues in Phd antitoxin buried inside the active site of Doc toxin attest to the likely function of this residue. We would consider it an important part of a thorough biochemical study of this archaeal PhD-Doc toxin antitoxin system, to be performed in the future.

- Lines 349 and 353 : same comment on the “homologs” term.

Response: Same as above, homology was defined based on PSI-BLAST.

REVIEWERS' COMMENTS

Reviewer #3 (Remarks to the Author):

We thank the reviewers for their response which was very clear and answered our comments. Congratulations on the great study.

Reviewer #5 (Remarks to the Author):

I co-reviewed this manuscript with one of the reviewers who provided the listed reports.